# TimeXL: Explainable Multi-modal Time Series Prediction with LLM-in-the-Loop

**Yushan Jiang**[1]*, **Wenchao Yu**[2]†, **Geon Lee**[3], **Dongjin Song**[1]†, **Kijung Shin**[3],
**Wei Cheng**[2], **Yanchi Liu**[2], **Haifeng Chen**[2]
[1]School of Computing, University of Connecticut
[2]Data Science & System Security Department, NEC Labs America
[3]Kim Jaechul Graduate School of AI, KAIST

## Abstract

Time series analysis provides essential insights for real-world system dynamics and informs downstream decision-making, yet most existing methods often overlook the rich contextual signals present in auxiliary modalities. To bridge this gap, we introduce TimeXL, a multi-modal prediction framework that integrates a prototype-based time series encoder with three collaborating Large Language Models (LLMs) to deliver more accurate predictions and interpretable explanations. First, a multi-modal prototype-based encoder processes both time series and textual inputs to generate preliminary forecasts alongside case-based rationales. These outputs then feed into a prediction LLM, which refines the forecasts by reasoning over the encoder's predictions and explanations. Next, a reflection LLM compares the predicted values against the ground truth, identifying textual inconsistencies or noise. Guided by this feedback, a refinement LLM iteratively enhances text quality and triggers encoder retraining. This closed-loop workflow—prediction, critique (reflect), and refinement—continuously boosts the framework's performance and interpretability. Empirical evaluations on four real-world datasets demonstrate that TimeXL achieves up to 8.9% improvement in AUC and produces human-centric, multi-modal explanations, highlighting the power of LLM-driven reasoning for time series prediction.

## 1 Introduction

In the modern big-data era, time series analysis has become indispensable for understanding real-world system behaviors and guiding downstream decision-making tasks across numerous domains, including healthcare, traffic, finance, and weather [1, 2, 3, 4]. Although deep learning models have demonstrated success in capturing complex temporal dependencies [5, 6, 7, 8], real-world time series are frequently influenced by external information beyond purely temporal factors. Such additional context, which may come from textual narratives (*e.g.*, finance news [9] or medical reports [10]), can offer critical insights for more accurate forecasting and explainability.

Recent multi-modal approaches for time series have shown promise by integrating rich contextual signals from disparate data sources, thereby improving performance across a range of tasks including forecasting, classification, imputation, and retrieval [11, 12, 13, 14]. While these approaches utilize supplementary data to enhance predictive accuracy, they often lack explicit mechanisms to systematically reason and explain about *why* or *how* contextual signals affect outcomes. This gap in interpretability poses significant barriers for high-stakes applications such as finance and healthcare, where trust and transparency are paramount.

---

*Most of this work was done during the internship at NEC Labs America.
†Correspondence to: Wenchao Yu <wyu@nec-labs.com>, Dongjin Song <dongjin.song@uconn.edu>.

39th Conference on Neural Information Processing Systems (NeurIPS 2025).

Meanwhile, Large Language Models (LLMs) [15, 16] have risen to prominence for their remarkable ability to process and reason over textual data across domains, enabling tasks like sentiment analysis, question answering, and content generation in zero- and few-shot settings [17, 18, 19]. Recent advancements in agentic designs further augment LLM capabilities, facilitating structured reasoning and iterative decision-making for context-rich scenarios [20, 21]. Their encoded domain knowledge makes them natural candidates for supporting real-world multi-modal time series analyses, where textual context (*e.g.*, news or expert notes) plays a vital role [22, 23, 24].

Motivated by these observations, we introduce TimeXL, a novel framework that adopts a closed-loop workflow of prediction, critique (reflect), and refinement, and unifies a prototype-driven time series encoder with LLM-based reasoning to deliver both accurate and interpretable multi-modal forecasting (Figure 1). Our approach first employs a *multi-modal prototype-based encoder* to generate preliminary time series predictions alongside human-readable explanations, leveraging case-based reasoning [25, 26, 27] from both the temporal and textual modalities. These explanations not only justify the encoder's predictions but also serve as auxiliary signals to guide an LLM-powered component that further refines the forecasts and contextual rationales.

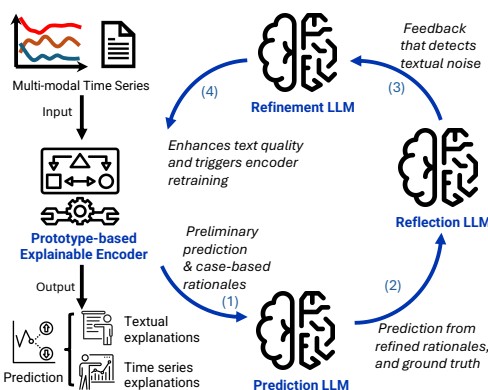

Unlike conventional methods that merely fuse multi-modal inputs for better accuracy, TimeXL iterates between predictive and refinement phases to mitigate textual noise, fill knowledge gaps, and produce more faithful explanations. Specifically, a *reflection LLM* diagnoses potential weaknesses by comparing predictions with ground-truth signals, while a *refinement LLM* incorporates these insights to update textual inputs and prototypes iteratively. This feedback loop progressively im-

Figure 1: An overview of the TimeXL workflow. A prototype-based explainable encoder first produces predictions and case-based rationales for both time series and text. The prediction LLM refines forecasts based on these rationales (Step 1). A reflection LLM then critiques the output against ground truth (Step 2), providing feedback to detect textual noise (Step 3). Finally, a refinement LLM updates the text accordingly, triggering encoder retraining for improved accuracy and explanations (Step 4). Note that the predictions from the encoder and the LLM are also fused to enhance overall accuracy.

proves both the predictive and explanatory capabilities of the entire system. Our contributions are summarized as follows:

- We present a prototype-based encoder that combines time series data with textual context, producing transparent, case-based rationales.

- We exploit the interpretative prowess of LLMs to reason over the encoder's outputs and iteratively refine both predictions and text, leading to improved prediction accuracy and explanations.

- Experiments on four real-world benchmarks show that TimeXL consistently outperforms baselines, achieving up to a 8.9% improvement in AUC while providing faithful, human-centric multi-modal explanations. Overall, TimeXL opens new avenues for explainable multi-modal time series analysis by coupling prototype-based inference with LLM-driven reasoning.

## 2    Related Work

Our work is primarily related to three key areas of time series research: multi-modality, explanation, and the use of LLMs. A more detailed discussion of related methods is provided in Appendix H.

**Multi-modal Time Series Analysis.** In recent years, multi-modal time series analysis has gained increasing attention in diverse domains such as finance, healthcare and environmental sciences [28, 11, 13]. Multiple approaches have been proposed to model interactions across different modalities, including fusion and alignment through concatenation [29], attention mechanism [12, 30, 31], gating functions [32], and contrastive learning [33, 34]. These methods facilitate tasks beyond forecasting

and classification, such as retrieval [14], imputation [13], and causal discovery [33]. Recent research efforts have also advanced the field through comprehensive benchmarks and studies [35, 36, 37], highlighting the incorporation of new modalities to boost task performance. Meanwhile, others explore transforming time series to other modalities [38, 39, 40, 41, 42] or generating synthetic ones [32, 43] for effective analysis. Nevertheless, these studies primarily focus on improving numerical performance while the deeper and tractable rationale behind *why* or *how* the textual or other contextual signals influence time series outcomes remains underexplored.

**Time Series Explanation.** Recent studies have explored diverse paradigms for time series interpretability. Gradient-based and perturbation-based saliency methods, for example, highlight important features at different time steps [44, 45], where some works explicitly incorporate temporal structures into models and objectives [46, 47]. Surrogate approaches also offer global or local explanations, such as applying Shapley values to time series [48], enforcing model consistency via self-supervised objectives [49], or using information-theoretic strategies for coherent explanations [50]. In contrast to saliency or surrogate-based explanations, we adopt a *case-based reasoning* paradigm [25, 26, 27], which end-to-end generates predictions and built-in explanations from learned prototypes. Our work extends this approach to multi-modal time series by producing human-readable reasoning artifacts for both the temporal and contextual modalities.

**LLMs for Time Series Analysis.** The rapid development of LLMs [15, 16] has begun to inspire new directions in time series research [51, 52]. Many existing methods fine-tune pre-trained LLMs on time series tasks and achieves promising results [53, 54, 55], where textual data (*e.g.*, domain instructions, metadata, summaries) are often encoded as prefix embeddings to enrich time series representations [56, 57, 58, 59]. These techniques also contribute to the emergence of time series foundation models [55, 60, 61, 62]. An alternative line of research leverages the zero-shot or few-shot capabilities of LLMs by directly prompting pre-trained language models with text-converted time series [63] or context-laden prompts representing domain knowledge [24], often yielding surprisingly strong performance in real-world scenarios. Furthermore, LLMs can act as knowledge inference modules, synthesizing high-level patterns or explanations that augment standard time series pipelines [64, 65, 66]. Building on this trend, recent works have explored time series reasoning with LLMs by framing tasks as natural-language problems, including time series understanding & question answering [67, 68, 69, 70, 71, 72], and language-guided inference [67, 69]. Compared with existing works, our work provides a unique synergy between an explainable model and LLMs, enhancing explainable prediction through iterative, grounded, and reflective interaction.

## 3 Methodology

In this section, we present the framework for explainable multi-modal time series prediction with LLMs. We first introduce the problem statement. Next, we present the design of a time series encoder that provides prediction and multi-modal explanations as the basis. Finally, we introduce three language agents interacting with the encoder towards better prediction and reasoning results.

### 3.1 Problem Statement

In this paper, we consider a multi-modal time series prediction problem. Each instance is represented by the multi-modal input $(\boldsymbol{x}, \boldsymbol{s})$, where $\boldsymbol{x} = (x_1, x_2, \cdots, x_T) \in \mathbb{R}^{N \times T}$ denotes time series data with $N$ variables and $T$ historical time steps, and $\boldsymbol{s}$ denotes the corresponding text data describing the real-world context. The text data $\boldsymbol{s}$ can be further divided into $L$ meaningful segments. Based on the historical time series and textual context, our objective is to predict the future outcome $\boldsymbol{y}$, either as a discrete value for classification tasks, or as a continuous value for regression tasks. We focus on the classification task to reflect discrete decision-making scenarios common in real-world multi-modal applications, where interpretability is often essential for reliable decision support. We also include an evaluation of the regression setting in the Appendix G. There are four major components in the proposed TimeXL framework, a multi-modal prototype encoder $\mathcal{M}_{\mathrm{enc}}$ that provides an initial prediction and case-based explanations, a prediction LLM $\mathcal{M}_{\mathrm{pred}}$ that provides a prediction based on contextual understanding with explanations, a reflection LLM $\mathcal{M}_{\mathrm{refl}}$ that generates feedback, and a refinement LLM $\mathcal{M}_{\mathrm{refine}}$ that refines the textual context based on the feedback. Below, we introduce each component and how they synergize toward better prediction and explanation.

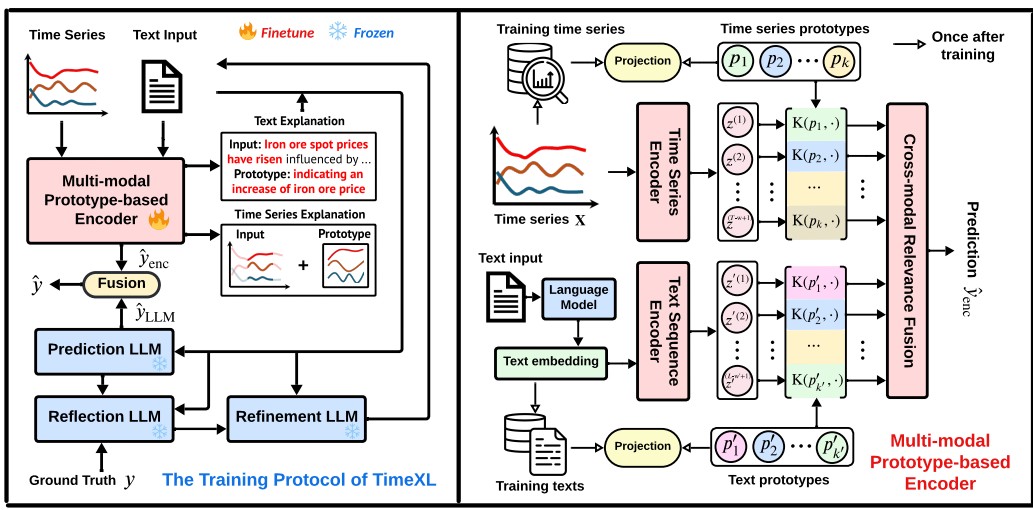

Figure 2: The training protocol of TimeXL (left), and multi-modal prototype-based encoder (right).

## 3.2 Multi-modal Prototype-based Encoder

We design a multi-modal prototype-based encoder that can generate predictions and explanations across different modalities in an end-to-end manner, as shown in Figure 2. We introduce the model architecture, the learning objectives that yield good explanation properties of prototypes, and the pipeline of case-based explanations using prototypes.

### 3.2.1 Multi-modal Sequence Modeling with Prototypes

**Sequence Encoder.** To capture both temporal and semantic dependencies, we adopt separate encoders for time series ($\mathcal{E}_\theta$) and text ($\mathcal{E}_\phi$). For $x \in \mathbb{R}^{N \times T}$, the time series encoder $\mathcal{E}_\theta$ maps the entire sequence into one or multiple representations, which serve as candidates for prototype learning. Simultaneously, the text input $s$ is first transformed by a *frozen* pre-trained language model, PLM (*e.g.*, BERT [73] or Sentence-BERT [74]), to produce embeddings $e_s \in \mathbb{R}^{d_s \times L}$. These embeddings are then processed by a separate encoder $\mathcal{E}_\phi$ to extract meaningful text features. It is worth noting that the choice of $\mathcal{E}_\theta$ and $\mathcal{E}_\phi$ also affects the granularity of explanations. As we will introduce shortly, the prototypes are learned based on sequence representations and are associated with the counterparts in the input space, where the correspondences are determined by the encoders. In this paper, we choose convolution-based encoders for both modalities to capture the fine-grained sub-sequence (*i.e.,* segment) patterns:

$$\boldsymbol{Z}_{\text{time}} = \big(\boldsymbol{z}_1, \ldots, \boldsymbol{z}_{T-w+1}\big) \ = \ \mathcal{E}_\theta(\boldsymbol{x}), \quad \boldsymbol{Z}_{\text{text}} = \big(\boldsymbol{z}_1', \ldots, \boldsymbol{z}_{L-w'+1}'\big) \ = \ \mathcal{E}_\phi(\boldsymbol{e}_s) \qquad (1)$$

where $\boldsymbol{z}_i \in \mathbb{R}^h$ and $\boldsymbol{z}_j' \in \mathbb{R}^{h'}$ denote segment-level representations learned via convolutional kernels of sizes $w$ and $w'$, respectively.

**Prototype Allocation.** To establish interpretability, we learn a set of *time series prototypes* and *text prototypes* for each class $c \in \{1, \ldots, C\}$. Specifically, we introduce: $\boldsymbol{P}_{\text{time}}^{(c)} \in \mathbb{R}^{k \times h}, \boldsymbol{P}_{\text{text}}^{(c)} \in \mathbb{R}^{k' \times h'}$ so that each prototype $\boldsymbol{p}_i^{(c)} \in \mathbb{R}^h$ (time series) or $\boldsymbol{p}_i'^{(c)} \in \mathbb{R}^{h'}$ (text) resides in the same feature space as the relevant encoder outputs. For an input sequence, we measure the similarity between each prototype and the most relevant segment in the corresponding modality:

$$\text{Sim}_i^{(c)} = \max(\text{Sim}_{i,1}^{(c)}, \cdots, \text{Sim}_{i,T-w+1}^{(c)}), \text{where} \ \ \text{Sim}_{i,j}^{(c)} = \exp\left(-\left\|\boldsymbol{p}_i^{(c)} - \boldsymbol{z}_j\right\|_2^2\right) \in [0, 1] \quad (2)$$

We aggregate similarity scores across all prototypes for each modality, yielding $\text{Sim}_{\text{time}} \in \mathbb{R}^{kC}$ and $\text{Sim}_{\text{text}} \in \mathbb{R}^{k'C}$. Finally, we jointly consider the cross-modal relevance and use a non-negative fusion weight matrix $\boldsymbol{W} \in \mathbb{R}^{C \times (k+k')}$ that translates these scores into class probabilities:

$$\hat{\boldsymbol{y}}_{\text{enc}} \ = \ \text{Softmax}\Big(\boldsymbol{W}\big[\text{Sim}_{\text{time}} \| \text{Sim}_{\text{text}}\big]\Big) \in [0, 1]^C. \qquad (3)$$

### 3.2.2 Learning Prototypes toward Better Explanation

**Learning Objectives.** The learning objectives include three regularization terms that reinforce the interpretability of multi-modal prototypes. In this paper, we focus on a predicting discrete label, where the basic objective is the cross-entropy loss for the prediction drawn from multi-modal explainable artifacts $\mathcal{L}_{\mathrm{CE}} = \sum_{\boldsymbol{x},\boldsymbol{s},\boldsymbol{y}} \boldsymbol{y} \log(\hat{\boldsymbol{y}}_{\mathrm{enc}}) + (1-\boldsymbol{y})\log(1-\hat{\boldsymbol{y}}_{\mathrm{enc}})$. Besides, we encourage a clustering structure of segments in the representation space by enforcing each segment representation to be adjacent to its closest prototype. Reversely, we regularize each prototype to be as close to a segment representation as possible, to help the prototype locate the most evidencing segment. Both regularization terms are denoted as $\mathcal{L}_c$ and $\mathcal{L}_e$, respectively, where we omit the modality and class notations for ease of understanding:

$$\mathcal{L}_c = \sum_{\boldsymbol{z}_j \in \boldsymbol{Z}_{(\cdot)}} \min_{\boldsymbol{p}_i \in \boldsymbol{P}_{(\cdot)}} \|\boldsymbol{z}_j - \boldsymbol{p}_i\|_2^2, \quad \mathcal{L}_e = \sum_{\boldsymbol{p}_i \in \boldsymbol{P}_{(\cdot)}} \min_{\boldsymbol{z}_j \in \boldsymbol{Z}_{(\cdot)}} \|\boldsymbol{p}_i - \boldsymbol{z}_j\|_2^2 \tag{4}$$

Moreover, we encourage a diverse structure of prototype representations to avoid redundancy and maintain a compact explanation space, by penalizing their similarities via a hinge loss $\mathcal{L}_d$, with a threshold $d_{\min}$ :

$$\mathcal{L}_d = \sum_{i=1} \sum_{j \neq i} \max\left(0, d_{\min} - \|\boldsymbol{p}_i - \boldsymbol{p}_j\|_2^2\right) \tag{5}$$

The full objective is written as: $\mathcal{L} = \mathcal{L}_{\mathrm{CE}} + \lambda_1 \mathcal{L}_c + \lambda_2 \mathcal{L}_e + \lambda_3 \mathcal{L}_d$, with hyperparameters $\lambda_1, \lambda_2$, and $\lambda_3$ that balance regularization terms towards achieving an optimal and explainable prediction.

**Prototype Projection.** After learning objectives converge, the multi-modal prototypes are well-regularized and reflect good explanation properties. However, these prototypes are still not readily explainable as they are only close to some exemplar segments in the representation space. Therefore, we perform prototype projection to associate each prototype with a training segment from its own class that preserves $\mathcal{L}_e$ in the representation space, for both time series and text:

$$\boldsymbol{p}_i^{(c)} \leftarrow \underset{\boldsymbol{z}_j \in \boldsymbol{Z}_{(\cdot)}^{(c)}}{\arg\min} \left\|\boldsymbol{p}_i^{(c)} - \boldsymbol{z}_j\right\|_2^2, \quad \forall \boldsymbol{p}_i^{(c)} \in \boldsymbol{P}_{(\cdot)}^{(c)} \tag{6}$$

By associating each prototype with a training segment in the representation space, the multi-modal physical meaning is induced. During testing phase, a multi-modal instance will be compared with prototypes across different modalities to infer predictions, where the similarity scores and prototypes' class information assemble the explanation artifacts for reasoning.

## 3.3 Explainable Prediction with LLM-in-the-Loop

To further leverage the reasoning and inference capabilities of LLMs in real-world time series contexts, we propose a framework with three interacting LLM agents: a prediction agent $\mathcal{M}_{\mathrm{pred}}$, a reflection agent $\mathcal{M}_{\mathrm{refl}}$, and a refinement agent $\mathcal{M}_{\mathrm{refine}}$. These LLM agents interact with the multi-modal prototype-based encoder $\mathcal{M}_{\mathrm{enc}}$ toward better prediction accuracy and explainability.

### 3.3.1 Model Synergy for Augmented Prediction

**Prediction with Enriched Contexts.** The prediction LLM agent $\mathcal{M}_{\mathrm{pred}}$ generates predictions based on the input text $\boldsymbol{s}$. To improve prediction accuracy, the encoder $\mathcal{M}_{\mathrm{enc}}$ supplements $\boldsymbol{s}$ with *case-based explanations*. Specifically, $\mathcal{M}_{\mathrm{enc}}$ selects the $\omega$ prototypes that exhibit the highest relevance to any of the textual segments within $\boldsymbol{s}$. Relevance is determined by the similarity scores used in Equation 2. These selected prototypes are then added to the input prompt of $\mathcal{M}_{\mathrm{pred}}$ as explanations, providing richer real-world context and leading to more accurate predictions. The $\omega$ prototype-segment pairs, which construct the explanation $\mathbf{expl}_s$ of the input text $\boldsymbol{s}$, are retrieved as follows:

$$\mathbf{expl}_s = \left\{\left(\boldsymbol{p}_i'^{(c)}, \boldsymbol{s}_j\right) : (i,j,c) \in \text{Top-}\omega(\mathrm{Sim}_{\mathrm{text}})\right\}, \text{Top-}\omega(\mathrm{Sim}_{\mathrm{text}}) = \arg\text{Top-}\omega_{(i,j,c)}\left(\mathrm{Sim}_{i,j}'^{(c)}\right)$$

Note that here $\boldsymbol{p}_i'^{(c)}$ denotes a raw text segment, and $i,j,c$ denote the prototype, segment, and class indices, respectively. As $\mathbf{expl}_s$ contains relevant contextual guidance, it augments the input space and removes semantic ambiguity for prediction agent $\mathcal{M}_{\mathrm{pred}}$. Therefore, the prediction is drawn as $\hat{\boldsymbol{y}}_{\mathrm{LLM}} = \mathcal{M}_{\mathrm{pred}}(\boldsymbol{s}, \mathbf{expl}_s)$. The prompt for querying $\mathcal{M}_{\mathrm{pred}}$ is provided in Appendix E, Figure 14.

**Fused Predictions.** We compile the final prediction based on a fusion of both the multi-modal encoder $\mathcal{M}_{\text{enc}}$ and prediction LLM $\mathcal{M}_{\text{pred}}$. Specifically, we linearly combine the continuous prediction probabilities $\hat{\boldsymbol{y}}_{\text{enc}}$ and discrete prediction $\hat{\boldsymbol{y}}_{\text{LLM}}$: $\hat{\boldsymbol{y}} = \alpha\hat{\boldsymbol{y}}_{\text{enc}} + (1 - \alpha)\hat{\boldsymbol{y}}_{\text{LLM}}$, where $\alpha \in [0, 1]$ is the hyperparameter selected from validation data. The encoder $\mathcal{M}_{\text{enc}}$ and prediction agent $\mathcal{M}_{\text{pred}}$ enhance each other based on their unique strengths. The $\mathcal{M}_{\text{enc}}$ is fine-tuned based on explicit supervised signals, ensuring accuracy in capturing temporal and contextual dependencies of multi-modal time series. On the other hand, $\mathcal{M}_{\text{pred}}$ contributes deep semantic understanding drawn from extensive text corpora. By fusing predictions from two distinct perspectives, we achieve a synergistic augmentation toward more accurate predictions for complex multi-modal time series.

### 3.3.2 Iterative Context Refinement via Reflective Feedback

While the prediction agent $\mathcal{M}_{\text{pred}}$ leverages the explainable artifacts to make informed predictions, it is not inherently designed to fit into the context of multi-modal time series data, which could lead to inaccurate predictions when the quality of textual context is inferior. To tackle this issue, we exploit another two language agents $\mathcal{M}_{\text{refl}}$ and $\mathcal{M}_{\text{refine}}$ to generate reflective feedback and refinements on the context, respectively, for better predictive insights.

Given the prediction $\hat{\boldsymbol{y}}_{\text{LLM}}$ generated by the prediction agent $\mathcal{M}_{\text{pred}}$, the reflection agent $\mathcal{M}_{\text{refl}}$ aims to understand the reasoning behind the implicit prediction logic of $\mathcal{M}_{\text{pred}}$. Specifically, it generates a *reflective feedback*, Refl, by analyzing the input text $\boldsymbol{s}$ and its prediction $\hat{\boldsymbol{y}}_{\text{LLM}}$, against the ground truth $\boldsymbol{y}$, to provide actionable insights for refinement, *i.e.,* $\text{Refl} = \mathcal{M}_{\text{refl}}(\boldsymbol{y}, \hat{\boldsymbol{y}}_{\text{LLM}}, \boldsymbol{s})$. Guided by the feedback, the refinement agent $\mathcal{M}_{\text{refine}}$ refines the previous text $\boldsymbol{s}_i$ into $\boldsymbol{s}_{i+1}$ by selecting and emphasizing the most relevant content, ensuring that important patterns are appropriately contextualized, which is similar to how a domain expert would perform, *i.e.,* $\boldsymbol{s}_{i+1} = \mathcal{M}_{\text{refine}}(\text{Refl}, \boldsymbol{s}_i)$. The prompts for querying $\mathcal{M}_{\text{refl}}$ and $\mathcal{M}_{\text{refine}}$ are provided in Figures 16, 17 18, 19, and discussed in Appendix E.

We finally integrate the refinement via reflection into the optimization loop of our proposed TimeXL, which is summarized in Algorithm 1. Once the textual context is improved, it is used to retrain the multi-modal prototype-based encoder $\mathcal{M}_{\text{enc}}$ for the next iteration. As such, the explanation (*e.g.,* quality of the prototypes) and predictive performance of $\mathcal{M}_{\text{enc}}$ can be improved through this iterative process. Consequently, the prediction agent $\mathcal{M}_{\text{pred}}$ could yield better prediction with more informative inputs, further enhancing the accuracy of $\hat{\boldsymbol{y}}$. We evaluate the predictive performance of LLM and save the encoder and reflection when an improvement is observed (Eval($\cdot$) pass) when applied to the validation set. In the testing phase, we select the reflections with the best validation performance to guide $\mathcal{M}_{\text{refine}}$ in context refinement, mimicking how an optimized deep learning model is applied to unseen testing data.

---

**Algorithm 1** Iterative Optimization Loop of TimeXL

**Inputs:** Multi-modal time series $(\boldsymbol{x}, \boldsymbol{s}, \boldsymbol{y})$ with $\mathcal{D}_{\text{train}}$, $\mathcal{D}_{\text{val}}$, $\mathcal{D}_{\text{test}}$, prototype-based encoder $\mathcal{M}_{\text{enc}}$, prediction agent $\mathcal{M}_{\text{pred}}$, reflection agent $\mathcal{M}_{\text{refl}}$, refinement agent $\mathcal{M}_{\text{refine}}$, fusion parameter $\alpha$, max iteration $\tau$, improvement check Eval($\cdot$)

**Training & Validation:**
Initialize $\boldsymbol{s}_0 = \boldsymbol{s}$, $i = 0$, $\mathcal{M}_{\text{enc}}^* \leftarrow \varnothing$, $\text{Refl}^* \leftarrow \varnothing$
**while** $i < \tau$ **do**
    Train $\mathcal{M}_{\text{enc}}$ using $\mathcal{D}_{\text{train,val}} = \{(\boldsymbol{x}, \boldsymbol{s}_i, \boldsymbol{y}), \cdots\}$
    $\hat{\boldsymbol{y}}_{\text{enc}}, \mathbf{expl}_{\boldsymbol{s}_i} = \mathcal{M}_{\text{enc}}(\boldsymbol{x}, \boldsymbol{s}_i)$
    $\hat{\boldsymbol{y}}_{\text{LLM}} = \mathcal{M}_{\text{pred}}(\boldsymbol{s}_i, \mathbf{expl}_{\boldsymbol{s}_i})$
    $\hat{\boldsymbol{y}} = \alpha\hat{\boldsymbol{y}}_{\text{enc}} + (1 - \alpha)\hat{\boldsymbol{y}}_{\text{LLM}}$
    **if** Eval(Refl, $\mathcal{D}_{\text{val}}$) *pass* **then**
        $\mathcal{M}_{\text{enc}}^* \leftarrow \mathcal{M}_{\text{enc}}$ , $\text{Refl}^* \leftarrow \text{Refl}$
    $\text{Refl} = \mathcal{M}_{\text{refl}}(\boldsymbol{y}, \hat{\boldsymbol{y}}_{\text{LLM}}, \boldsymbol{s}_i)$
    $\boldsymbol{s}_{i+1} = \mathcal{M}_{\text{refine}}(\text{Refl}, \boldsymbol{s}_i)$
    increment $i$
**return** $\mathcal{M}_{\text{enc}}^*$, $\text{Refl}^*$

**Testing:**
$\boldsymbol{s}' = \mathcal{M}_{\text{refine}}(\text{Refl}^*, \boldsymbol{s})$ for $\mathcal{D}_{\text{test}}$
$\hat{\boldsymbol{y}}_{\text{enc}}, \mathbf{expl}_{\boldsymbol{s}'} = \mathcal{M}_{\text{enc}}^*(\boldsymbol{x}, \boldsymbol{s}')$
$\hat{\boldsymbol{y}}_{\text{LLM}} = \mathcal{M}_{\text{pred}}(\boldsymbol{s}', \mathbf{expl}_{\boldsymbol{s}'})$
$\hat{\boldsymbol{y}} = \alpha\hat{\boldsymbol{y}}_{\text{enc}} + (1 - \alpha)\hat{\boldsymbol{y}}_{\text{LLM}}$

---

## 4 Experiments

### 4.1 Experimental Setup

**Datasets.** We evaluate methods on four multi-modal time series datasets from three different real-world domains, including weather, finance, and healthcare. The detailed data statistics are summarized in Table 3 of Appendix A.1. The **Weather** dataset contains meteorological reports and the hourly time series records of temperature, humidity, air pressure, wind speed, and wind direction in New York City. The task is to predict if it will rain in the next 24 hours, given the last 24 hours of weather

Table 1: The F1 score (F1) and AUROC (AUC) for TimeXL and state-of-the-art baselines on multi-modal time series datasets.

| Datasets → Methods ↓ | Weather | | Finance | | Healthcare (TP) | | Healthcare (MT) | |
|---|---|---|---|---|---|---|---|---|
| | F1 | AUC | F1 | AUC | F1 | AUC | F1 | AUC |
| DLinear [75] | 0.540 | 0.660 | 0.255 | 0.485 | 0.393 | 0.500 | 0.419 | 0.388 |
| Autoformer [76] | 0.546 | 0.590 | 0.565 | 0.747 | 0.774 | 0.918 | 0.683 | 0.825 |
| Crossformer [77] | 0.500 | 0.594 | 0.571 | 0.775 | 0.924 | 0.984 | 0.737 | 0.913 |
| TimesNet [78] | 0.494 | 0.594 | 0.538 | 0.756 | 0.794 | 0.867 | 0.765 | 0.944 |
| iTransformer [79] | 0.541 | 0.650 | 0.600 | 0.783 | 0.861 | 0.931 | 0.791 | 0.963 |
| TSMixer [80] | 0.488 | 0.534 | 0.465 | 0.689 | 0.770 | 0.797 | 0.808 | 0.931 |
| TimeMixer [81] | 0.577 | 0.658 | 0.571 | 0.776 | 0.822 | 0.887 | 0.824 | 0.935 |
| FreTS [82] | 0.623 | 0.688 | 0.546 | 0.737 | 0.887 | 0.950 | 0.751 | 0.762 |
| PatchTST [5] | 0.592 | 0.675 | 0.604 | 0.795 | 0.841 | 0.934 | 0.695 | 0.928 |
| LLMTime [83] | 0.587 | 0.657 | 0.519 | 0.643 | 0.802 | 0.817 | 0.769 | 0.803 |
| PromptCast [63] | 0.499 | 0.365 | 0.418 | 0.607 | 0.727 | 0.768 | 0.696 | 0.871 |
| OFA [53] | 0.501 | 0.606 | 0.512 | 0.745 | 0.774 | 0.879 | 0.851 | 0.977 |
| FSCA [58] | 0.563 | 0.647 | 0.592 | 0.790 | 0.820 | 0.891 | 0.872 | 0.977 |
| Time-LLM [56] | 0.613 | 0.699 | 0.589 | 0.792 | 0.671 | 0.864 | 0.733 | 0.912 |
| TimeCMA [57] | 0.636 | 0.731 | 0.559 | 0.727 | 0.729 | 0.828 | 0.693 | 0.843 |
| MM-iTransformer [35] | 0.608 | 0.689 | 0.605 | 0.793 | 0.926 | 0.986 | 0.901 | 0.990 |
| MM-PatchTST [35] | 0.621 | 0.718 | 0.619 | 0.812 | 0.863 | 0.968 | 0.780 | 0.929 |
| TimeCAP [65] | 0.668 | 0.742 | 0.611 | 0.801 | 0.954 | 0.983 | 0.942 | 0.988 |
| TimeXL | 0.696 | 0.808 | 0.631 | 0.797 | 0.987 | 0.996 | 0.956 | 0.997 |

records and summary. The **Finance** dataset contains the daily record of the raw material prices together with 14 related indices from January 2017 to July 2024. Given the last 5 business days of stock price data and news, the task is to predict if the target price will exhibit an increasing, decreasing, or neutral trend on the next business day. The **Healthcare** datasets include **Test-Positive (TP)** and **Mortality (MT)**, both comprising weekly records related to influenza activity. TP contains the number of respiratory specimens testing positive for Influenza A and B, while MT reports deaths caused by influenza and pneumonia. For both datasets, the task is to predict whether the respective target ratio, either the test-positive rate or the mortality ratio in the upcoming week will exceed the historical average, based on records and summaries from the previous 20 weeks.

**Baselines, Evaluation Metrics, and Setup** We compare TimeXL with state-of-the-art baselines for time series prediction, including Autoformer [76], Dlinear [75], Crossformer [77], TimesNet [78], PatchTST [5], iTransformer [79], FreTS [82], TSMixer [80], TimeMixer [81] (classification implemented by TSlib [78]) and LLM-based methods like LLMTime [83], PromptCast [63], OFA [53], Time-LLM [56], TimeCMA [57] and FSCA [58], where LLMTime and PromptCast don't need fine-tuning. While some methods are primarily used for time series prediction with continuous values, they can be easily adapted for discrete value prediction. We also evaluate the multi-modal time series methods. Besides the Time-LLM, TimeCMA and FSCA where input text is used for embedding re-programming and alignment, we also evaluate Multi-modal PatchTST and Multi-modal iTransformer from [35], as well as TimeCAP [65]. We evaluate the discrete prediction via F1 score and AUROC (AUC) score, due to label imbalance in real-world time series datasets. All datasets are split 6:2:2 for training, validation, and testing. All results are averaged over five experimental runs. We alternate different embedding methods for texts based on its average length (Sentence-BERT [74] for finance dataset and BERT [73] for the others). More details are provided in Appendix A.2, A.3, A.4.

## 4.2 Performance Evaluation

The results of predictive performance are shown in Table 1. It is notable that multi-modal methods generally outperform time series methods across all datasets. These methods include LLM methods (*e.g.*, Time-LLM [56], TimeCMA [57], FSCA [58]) that leverage text embeddings to enhance time series predictions. Moreover, the multi-modal variants (MM-iTransformer and MM-PatchTST [35]) improve the performance of state-of-the-art time series methods, suggesting the benefits of integrating

real-world contextual information. Besides, TimeCAP [65] integrates the predictions from both modalities, further improving the predictive performance. TimeXL constantly achieves the highest F1 and AUC scores, consistently surpassing both time series and multi-modal baselines by up to 8.9% of AUC (compared to TimeCAP on the Weather dataset). This underscores the advantage of TimeXL, which synergizes multi-modal encoder with language agents to enhance interpretability and thus predictive performance in multi-modal time series.

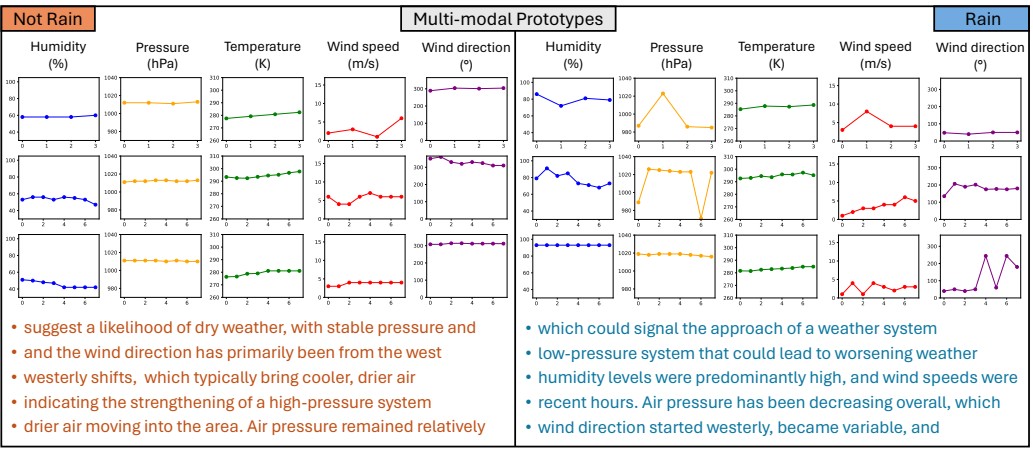

Figure 3: Key time series prototypes and text prototypes learned on the Weather dataset. Each row in the figure represents a time series prototype with different channels.

### 4.3 Explainable Multi-modal Prototypes

Next, we present the explainable multi-modal prototypes rendered by TimeXL, which establishes the case-based reasoning process. Figure 3 shows a subset of time series and text prototypes learned on the weather dataset. The time series prototypes demonstrate the typical temporal patterns aligned with different real-world weather conditions (*i.e.,* rain and not rain). For example, a constant or decreasing humidity at a moderate level, combined with high and steady air pressure, typically indicates a non-rainy scenario. The consistent wind direction is also a sign of mild weather conditions. On the contrary, high humidity, low and fluctuating pressure, along with variable winds typically reveal an unstable weather system ahead. In addition to time series, the text prototypes also highlight consistent semantic patterns for different weather conditions, such as the channel-specific (*e.g.*, drier air moving into the area, strengthening of high-pressure system) and overall (*e.g.*, a likelihood of dry weather) descriptions of weather activities. In Appendix D.1, we also present more multi-modal prototypes for the weather dataset in Figure 9, for the finance dataset in Figure 10, and for healthcare datasets in Figures 11 and 12. TimeXL provides coherent and informative prototypes from the exploitation of time series and its real-world contexts, which facilitates both prediction and explanation.

### 4.4 Multi-modal Case-based Reasoning

Building upon the multi-modal prototypes, we present a case study on the testing set of weather data, comparing the original and TimeXL's reasoning processes to highlight its explanatory capability, as shown in Figure 4. In this case, the original text is incorrectly predicted as not rain. We have three key observations: **(1)** The refinement process filters the original text to emphasize weather conditions more indicative of rain, guided by reflections from training examples. The refined text preserves the statement on stability while placing more emphasis on humidity and wind as key indicators. **(2)** Accordingly, the matched segment-prototype pairs from the original text focus more on temperature stability and typical diurnal variations, while the matched pairs in the refined text highlights wind variability, moisture transport, and approaching weather system, aligning more with rain conditions. **(3)** Furthermore, the reasoning on time series provides a complementary view for assessing weather conditions. The matched time series prototypes identify high humidity and its drop-and-rise trends, wind speed fluctuations and directional shifts, and the declining phase of air pressure fluctuations, all of which are linked to the upcoming rainy conditions. The matched multi-modal prototypes

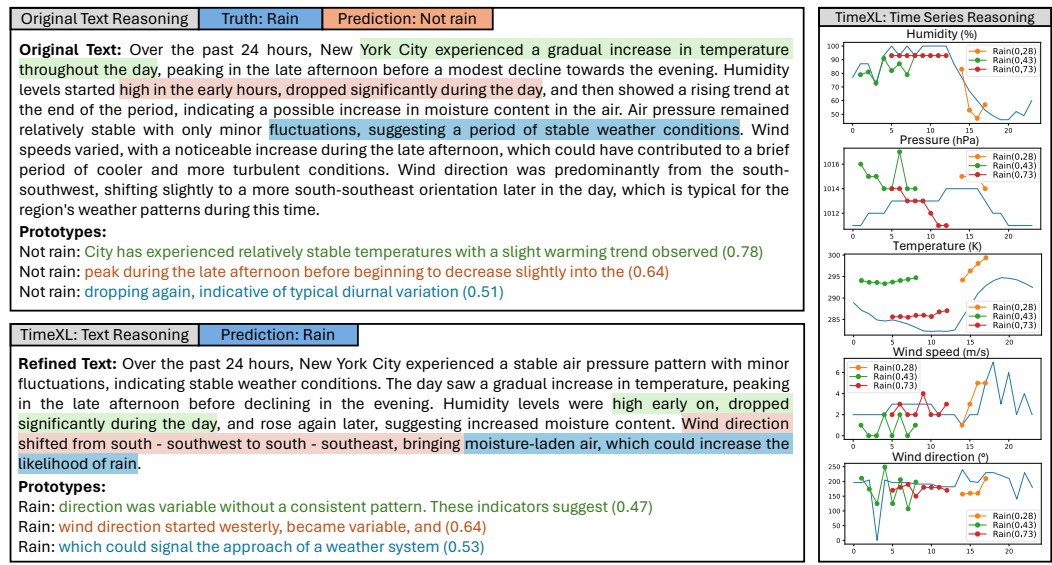

Figure 4: Multi-modal case-based reasoning example on the Weather dataset. The left part illustrates the reasoning process for both the original and refined text in TimeXL, with matched prototype-input pairs highlighted in the same color along with their similarity scores. The right part presents the time series reasoning in TimeXL, where matched prototypes are overlaid on the time series.

from TimeXL demonstrate its effectiveness in capturing relevant information for both predictive and explanatory analysis. We also provide a case study on the Finance dataset in Figure 13, where textual explanations are generated at the granularity of a half-sentence.

## 4.5 Iterative Analysis

To verify the effectiveness of overall workflow with reflection and refinement LLMs as shown in Figure 1, we conduct an iterative analysis of text quality and TimeXL performance, as shown in Figure 5. Specifically, we evaluate the text quality based on its zero-shot predictive accuracy using an LLM. Notably, the text quality benefits from iteration improvements and mostly saturates after one or two iterations. Correspondingly, TimeXL performance quickly improves and stabilizes with very minor fluctuations. These observations underscore how TimeXL alternates between predictive and reflective refinement phases to mitigate textual noise, thus enhancing its predictive capability.

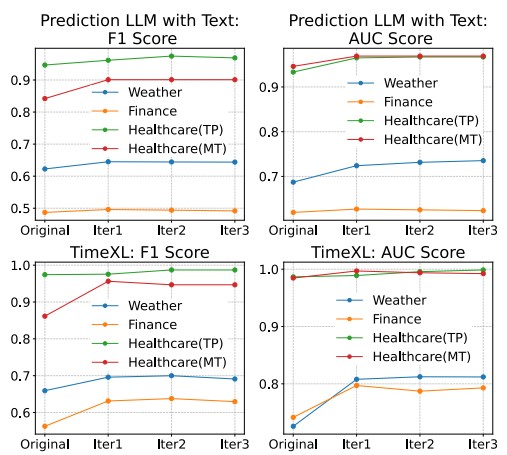

Figure 5: The text quality and TimeXL performance over iterations.

## 4.6 Ablation Studies

We present the component ablations of TimeXL in Table 2, where the texts are refined using the reflections with the best validation performance. Firstly, the multi-modal encoder consistently outperforms single-modality variants across all datasets, highlighting the benefit of integrating time series and textual modalities. Secondly, the prediction LLM exhibits better performance than text-only encoder, underscoring the advantage of LLM's contextual understanding and reasoning. Furthermore, the text prototypes consistently improve the predictive performance of LLM, underscoring the effectiveness of explainable artifacts from the multi-modal encoder, in terms of providing relevant contextual guidance. Finally, the fusion of prediction LLM and multi-modal encoder further enhances the predictive performance, surpassing the best results achieved by either component alone. These

Table 2: Ablation studies of TimeXL on multi-modal time series datasets.

| Ablation ↓ | Variants | Weather | | Finance | | Healthcare (Test-Positive) | | Healthcare (Mortality) | |
|---|---|---|---|---|---|---|---|---|---|
| | | F1 | AUC | F1 | AUC | F1 | AUC | F1 | AUC |
| Encoder | Time Series | 0.602 | 0.691 | 0.585 | 0.751 | 0.889 | 0.957 | 0.861 | 0.965 |
| | Text | 0.567 | 0.658 | 0.472 | 0.636 | 0.871 | 0.941 | 0.840 | 0.887 |
| | Multi-modal | 0.674 | 0.767 | 0.619 | 0.791 | 0.934 | 0.974 | 0.937 | 0.988 |
| LLM | Text | 0.645 | 0.724 | 0.496 | 0.627 | 0.974 | 0.967 | 0.901 | 0.969 |
| | Text + Prototype | 0.667 | 0.739 | 0.544 | 0.662 | 0.987 | 0.983 | 0.952 | 0.976 |
| Fusion | Select-Best | 0.674 | 0.767 | 0.619 | 0.791 | 0.987 | 0.983 | 0.952 | 0.988 |
| | TimeXL | **0.696** | **0.808** | **0.631** | **0.797** | **0.987** | **0.996** | **0.956** | **0.997** |

observations demonstrate the advantage of our framework synergizing the time series model and LLM for mutually augmented prediction. In Appendix B, additional ablation and model studies are presented, including analyses of the multi-modal encoder (Figure 6; Table 4, 5, 6), quantitative evaluation of case-based explanations (Figure 7), and comparisons of base LLMs (Table 7; Figure 8).

## 5 Conclusions

In this paper, we present TimeXL, an explainable multi-modal time series prediction framework that synergizes a designed prototype-based encoder with three collaborative LLM agents in the loop (prediction, reflection, and refinement) to deliver more accurate predictions and explanations. Experiments on four multi-modal time series datasets show the advantages of TimeXL over state-of-the-art baselines and its excellent explanation capabilities.

## 6 Acknowledgments

Dongjin Song gratefully acknowledges the support of the National Science Foundation under Grant No. 2338878, as well as the generous research gifts from NEC Labs America and Morgan Stanley.

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

# A  Experimental Settings

## A.1  Dataset Statistics

In this subsection, we provide more details of the real-world datasets we used for the experiments. The data statistics are summarized in Table 3, including the meta information (*e.g.*, domain resolution, duration of real-world time series records), the number of channels and timesteps and so on. We use the Weather and Healthcare datasets in TimeCAP [65], and a Finance dataset extended from [12].

The **Weather** dataset contains the hourly time series record of temperature, humidity, air pressure, wind speed, and wind direction[3], and related weather summaries in New York City from October 2012 to November 2017. The task is to predict if it will rain in the next 24 hours, given the last 24 hours of weather records and summary.

The **Finance** dataset contains the daily record of the raw material prices together with 14 related indices ranging from January 2017 to July 2024[4] with news articles. The task is to predict if future prices will increase by more than 1%, decrease by more than 1%, or exhibit a neutral trend on the next business day, given the last 5 business days of stock price data and news.

The Healthcare datasets are related to testing cases and deaths of influenza[5]. The **Healthcare (Test-Positive)** dataset consists of the weekly records of the number of positive specimens for Influenza A and B, and related healthcare reports. The task is to predict if the percentage of respiratory specimens testing positive in the upcoming week for influenza will exceed the average value, given the records and summary in the last 20 weeks. Similarly, the **Healthcare (Mortality)** dataset contains the weekly records and healthcare reports of influenza and pneumonia deaths. The task is to predict if the mortality ratio from influenza and pneumonia will exceed the average value, given the records and summary in the last 20 weeks.

Table 3: Summary of dataset statistics.

| Domain | Dataset | Resolution | # Channels | # Timesteps | Duration | Ground Truth Distribution |
|--------|---------|-----------|-----------|-------------|----------|---------------------------|
| Weather | New York | Hourly | 5 | 45,216 | 2012.10 - 2017.11 | Rain (24.26%) / Not rain (75.74%) |
| Finance | Raw Material | Daily | 15 | 1,876 | 2012.09 - 2022.02 | Inc. (36.7%) / Dec. (34.1%) / Neutral (29.2%) |
| Healthcare | Test-Positive | Weekly | 6 | 447 | 2015.10 - 2024.04 | Not exceed (65.77%) / Exceed (34.23%) |
| Healthcare | Mortality | Weekly | 4 | 395 | 2016.07 - 2024.06 | Not exceed (69.33%) / Exceed (30.67%) |

## A.2  Hyperparameters

First, we provide the hyperparameters of baseline methods. Unless otherwise specified, we used the default hyperparameters from the Time Series Library (TSLib) [78]. For LLMTime [83], OFA [53], Time-LLM [56], TimeCMA [57], TimeCAP [65], FSCA [58], we use their own implementations. For all methods, the dropout rate $\in \{0.0, 0.1, 0.2\}$, learning rate $\in \{0.0001, 0.0003, 0.001\}$. For transformer-based and LLM fine-tuning methods [76, 77, 79, 5, 57, 56], the number of attention layers $\in \{1, 2\}$, the number of attention heads $\in \{4, 8, 16\}$. For Dlinear [75], moving average $\in \{3, 5\}$. For TimesNet [78] the number of layers $\in \{1, 2\}$. For PatchTST [5], MM-PatchTST [35] and FSCA [58], the patch size $\in \{3, 5\}$ for the finance dataset. For TimeCAP [65], we use the encoder with the prediction LLM, given the inherent multi-modal time series input. Next, we provide the hyperparameters of TimeXL. The numbers of time series prototypes and text prototypes are $k \in \{5, 10, 15, 20\}$ and $k' \in [5, 10]$, respectively. The hyperparameters controlling regularization strengths are $\lambda_1$, $\lambda_2$, $\lambda_3 \in [0.1, 0.3]$ with interval 0.05 for individual modality, $d_{\min} \in \{1.0, 1.5, 2.0\}$ for time series, $d_{\min} \in \{3.0, 3.5, 4.0\}$ for text. Learning rate for multi-modal encoder $\in \{0.0001, 0.0003, 0.001\}$, using Adam [84] as the optimizer. The number of case-based explanations fed to prediction LLM $\omega \in \{3, 5, 8, 10\}$. All results are reported as the average of five experimental runs.

---

[3]https://www.kaggle.com/datasets/selfishgene/historical-hourly-weather-data
[4]https://www.indexmundi.com/commodities
[5]https://www.cdc.gov/fluview/overview/index.html

### A.3    Large Language Model

We employed the gpt-4o-2024-08-06 version for GPT-4o in OpenAI API by default. We use the parameters max_tokens=2048, top_p=1, and temperature=0.7 for content generation (self-reflection and text refinement), and 0.3 for prediction. We keep the same setting for Gemini-2.0-Flash and GPT-4o-mini due to the best empirical performance.

### A.4    Environment

We conducted all the experiments on a TensorEX server with 2 Intel Xeon Gold 5218R Processor (each with 20 Core), 512GB memory, and 4 RTX A6000 GPUs (each with 48 GB memory).

## B    More Ablation Studies and Model Analysis

**Ablations of learning objectives:** We provide an ablation study on the learning objectives of TimeXL encoder, as shown in Figure 6. The results clearly show that the full objective consistently achieves the best encoder prediction performance, highlighting the necessity of regularization terms that enhance the interpretability of multi-modal prototypes. The clustering ($\lambda_1$) and evidencing ($\lambda_2$) objectives also play a crucial role in accurate prediction: the clustering term ensures distinguishable prototypes across different classes, while the evidencing term ensures accurate projection onto training data.

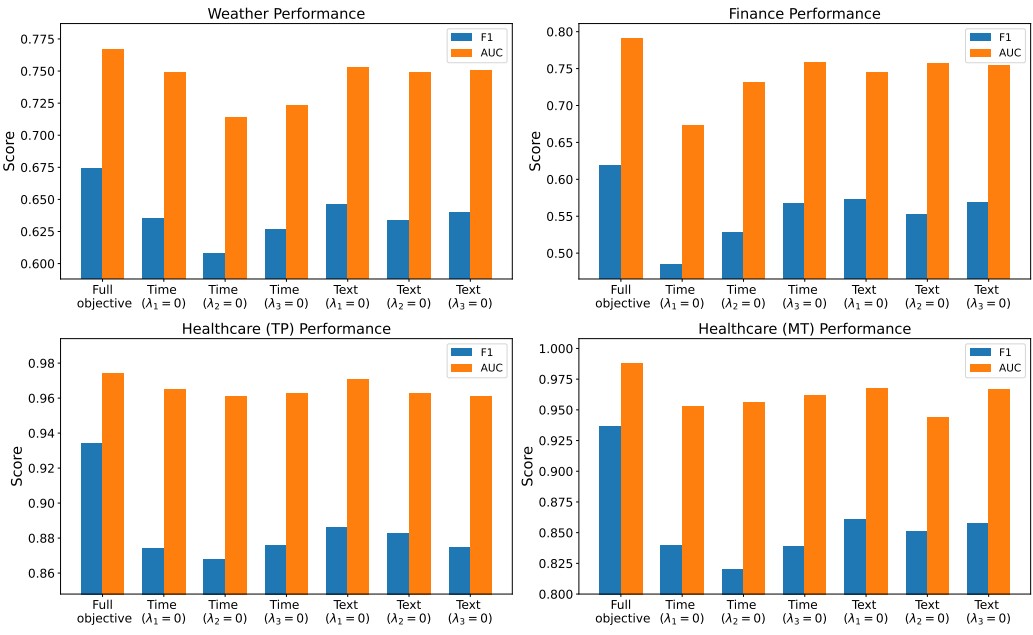

Figure 6: Ablation study of multi-modal encoder learning objectives.

**Sensitivity of regularization strengths:** To quantitatively evaluate the sensitivity of regularization strengths in the learning objectives, we report the performance of the multi-modal encoder on the Weather dataset (using the refined test texts), as detailed in Appendix A.2. As shown in Table 4, model performance remains relatively stable across different values. For both modalities, moderate values of $\lambda_1$ and $\lambda_2$ tend to yield better performance, suggesting the clustering and evidencing effectively guide meaningful prototype learning. A slight diversity constraint ($\lambda_3$) also helps to keep a compact interpretation prototype space.

**Sensitivity of prototype lengths:** We evaluate the sensitivity of the multi-modal encoder to prototype length by varying the single-kernel sizes for both time-series and text modalities on the Weather and Finance datasets (using the refined test texts), as shown in Table 5 and 6. We observe that performance is relatively stable across a range of prototype lengths on the Weather dataset, with optimal results

Table 4: Sensitivity analysis of different $\lambda$ values for multi-modal encoder on the Weather dataset.

| Modality | | | 0.1 | 0.15 | 0.2 | 0.25 | 0.3 |
|---|---|---|---|---|---|---|---|
| **Time** | $\lambda_1$ | F1 | 0.628±0.030 | 0.627±0.027 | 0.633±0.029 | 0.627±0.028 | 0.625±0.030 |
| | | AUC | 0.733±0.022 | 0.731±0.026 | 0.736±0.026 | 0.731±0.024 | 0.729±0.023 |
| | $\lambda_2$ | F1 | 0.632±0.028 | 0.634±0.028 | 0.627±0.032 | 0.626±0.028 | 0.628±0.028 |
| | | AUC | 0.734±0.024 | 0.736±0.022 | 0.733±0.028 | 0.729±0.021 | 0.733±0.025 |
| | $\lambda_3$ | F1 | 0.631±0.028 | 0.631±0.027 | 0.630±0.028 | 0.624±0.032 | 0.628±0.029 |
| | | AUC | 0.736±0.022 | 0.735±0.024 | 0.733±0.024 | 0.727±0.026 | 0.733±0.025 |
| **Text** | $\lambda_1$ | F1 | 0.629±0.027 | 0.629±0.026 | 0.631±0.027 | 0.635±0.027 | 0.634±0.028 |
| | | AUC | 0.732±0.024 | 0.736±0.024 | 0.740±0.026 | 0.738±0.023 | 0.735±0.025 |
| | $\lambda_2$ | F1 | 0.632±0.031 | 0.631±0.026 | 0.635±0.024 | 0.629±0.026 | 0.630±0.028 |
| | | AUC | 0.734±0.027 | 0.736±0.021 | 0.740±0.024 | 0.735±0.021 | 0.736±0.028 |
| | $\lambda_3$ | F1 | 0.635±0.026 | 0.633±0.023 | 0.628±0.025 | 0.625±0.029 | 0.627±0.027 |
| | | AUC | 0.740±0.021 | 0.736±0.025 | 0.731±0.023 | 0.729±0.027 | 0.731±0.026 |

typically achieved at moderate lengths for both modalities. For time series, the 8-hour segments are often sufficient to capture the typical temporal patterns of weather conditions, and a 12-token text window (with BERT) provides enough local contexts for weather prediction. For the Finance dataset, which operates on a business-day granularity, performance improves with longer lengths of time series prototypes that capture important financial trends. For the textual modality, we use Sentence-BERT to embed financial news at the half-sentence granularity, and shorter text windows (1-2 segments) typically provide compact and indicative market conditions. These results suggest that our model maintains robust to prototype length selection across a reasonable range, guided by the temporal characteristics of each dataset.

Table 5: Sensitivity analysis of prototype lengths for multi-modal encoder on the Weather dataset.

| Modality | Length | F1 | AUC |
|---|---|---|---|
| **Time** | 4 | 0.632±0.018 | 0.748±0.011 |
| | 8 | 0.642±0.011 | 0.754±0.008 |
| | 12 | 0.625±0.013 | 0.746±0.007 |
| | 16 | 0.621±0.012 | 0.743±0.007 |
| **Text** | 4 | 0.619±0.014 | 0.743±0.008 |
| | 8 | 0.626±0.015 | 0.744±0.008 |
| | 12 | 0.647±0.008 | 0.757±0.004 |
| | 16 | 0.627±0.009 | 0.748±0.007 |

Table 6: Sensitivity analysis of prototype lengths for multi-modal encoder on the Finance dataset.

| Modality | Length | F1 | AUC |
|---|---|---|---|
| **Time** | 1 | 0.430±0.002 | 0.595±0.002 |
| | 2 | 0.454±0.028 | 0.630±0.021 |
| | 3 | 0.475±0.006 | 0.662±0.006 |
| | 4 | 0.526±0.014 | 0.714±0.006 |
| | 5 | 0.588±0.029 | 0.769±0.018 |
| **Text** | 1 | 0.508±0.073 | 0.666±0.074 |
| | 2 | 0.496±0.061 | 0.673±0.067 |
| | 3 | 0.458±0.045 | 0.664±0.056 |

**Effect of case-based explanations on LLM predictions and other quality assessments:** Moreover, we assess how the number of allocated case-based explanations enhances the prediction LLM, as shown in Figure 7. We conduct experiments on the Weather and Finance datasets, demonstrating that incorporating more relevant case-based explanations consistently improves prediction performance. This trend highlights the effectiveness of explainable artifacts in providing meaningful contextual guidance. Importantly, this analysis also serves as a proxy for quantitatively evaluating explanation quality, where retrieving higher-quality case-based examples tends to yield greater performance gains. Beyond this quantitative assessment, alternative strategies can be employed to evaluate explanation quality more explicitly, for example, using LLM as a judge [85] to rate the relevance and helpfulness of explanations in a human-understandable manner, with respect to ground truth.

**Effect of different base LLMs:** To explore the effect of different base LLMs (Gemini-2.0 Flash and GPT-4o-mini), we provide the experiment results and analysis on the Healthcare (Test-Positive) dataset. For fair comparisons, we used the same prompts and ran the same number of iterations and followed the same settings detailed in Appendix A.2, A.3, and A.4. The results are shown in Table 7. It can be observed that GPT-4o clearly outperforms both GPT-4o-mini and Gemini-2.0-Flash, due to its better reasoning and contextual understanding capabilities

| Base LLM | F1 | AUC |
|---|---|---|
| GPT-4o-mini | 0.932 | 0.981 |
| Gemini-2.0-Flash | 0.937 | 0.983 |
| GPT-4o (default) | 0.987 | 0.996 |

Table 7: Performance of different base LLMs on Healthcare (Test-Positive).

(larger model size & pre-training corpus). Both GPT-4o-mini and Gemini-2.0-Flash are still competitive compared with baselines listed in Table 1. It reveals the impact of the LLM capability on the effectiveness of our framework, especially in tasks demanding real-world context understanding. We also provide the plot of iterative analysis in Figure 8, where we can also observe the performance improvement over iterations for different base LLMs.

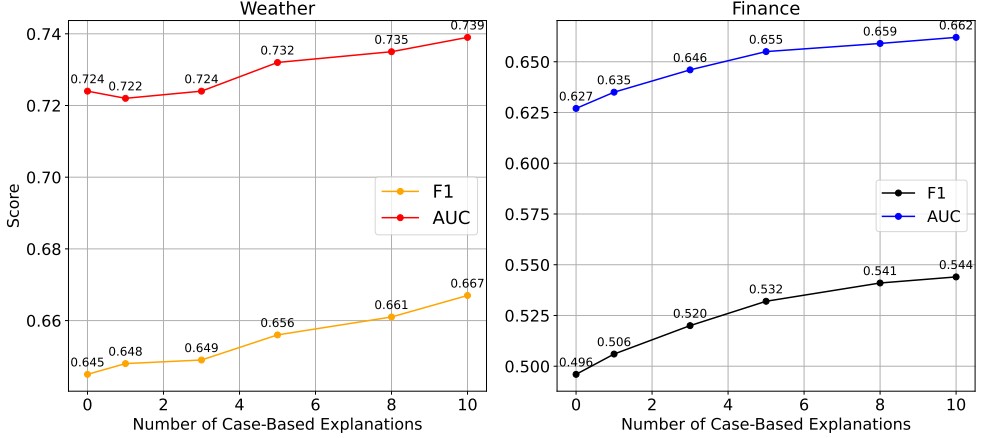

Figure 7: Effect of case-based explanations on LLM prediction performance.

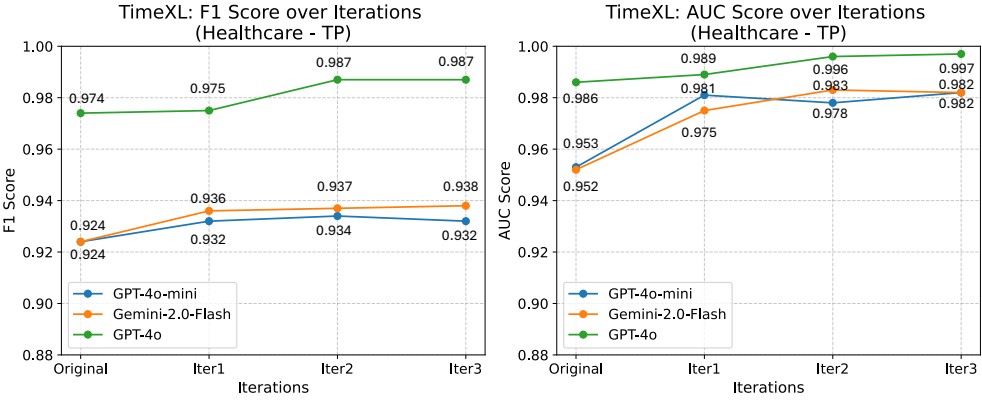

Figure 8: Iterative study of different base LLMs on the Healthcare (Test-Positive) dataset.

## C   Computation Cost for Iterative Optimization and Inference

We quantitatively analyze the token usage and runtime of the iterative process. Following the prompts and strategy detailed in Appendix E, each iteration over 4000 training and validation samples (average length of 400 tokens, around 300 words) requires about 9.74 M input tokens and yield 1.64 M output tokens. With GPT-4o (with input latency as TTFT $\approx 0.45$ seconds and output speed $\approx 156$ tokens/second), it corresponds to around 3.9 hours serially per iteration (prediction takes approximately 0.5 hours, reflection 0.08 hours, and refinement 3.35 hours). As prediction and refinement steps can also be parallelized, we can reduce the iteration time to about 12 minutes with 20 concurrent calls. Using a more efficient LLM base model Gemini-Flash-2.0 with TTFT $\approx 0.3$ seconds and $\approx 236$ output tokens/second, the iteration time can be further reduced to 8 minutes.

In the testing stage, we apply the fixed reflections selected during validation to refine the test texts. Based on our prompt lengths, the refinement and prediction steps can take down to 3.46 seconds per sample with GPT-4o, and 2.29 seconds per sample with Gemini-2.0-Flash, which is reasonable for real-world deployment.

In addition to efficient LLM backbones and parallelized querying, we discuss further strategies to lower computational cost. As shown in Figure 5, 1-2 iterations often yield clear performance gains. In practice, the iteration loop can be adaptively controlled by monitoring validation improvements and stopping early once gains fall below a predefined threshold. To further reduce the cost, we can also do selective refinement by skipping training examples that were already predicted accurately in previous iterations. These strategies help make our method more computationally tractable.

# D  Explainable Multi-modal Prototypes and Case Study

## D.1  Multi-modal Prototypes for All Datasets

We present the learned multi-modal prototypes across datasets including Weather (Figure 9), Finance (Figure 10), Healthcare (Test-Positive) (Figure 11), and Healthcare (Mortality) (Figure 12). It is noticeable that the prototypes from both modalities align well with real-world ground truth scenarios, ensuring faithful explanations and enhancing LLM predictions.

## D.2  Case-based Reasoning Example on the Finance Dataset

We provide another case-based reasoning example to demonstrate the effectiveness of TimeXL in explanatory analysis, as shown in Figure 13. In this example, the original text is incorrectly predicted as neutral instead of a decreasing trend of raw material prices. We have a few key observations based on the results. First, the refinement LLM filters the original text to emphasize economic and market conditions more indicative of a declining trend, based on the reflections from training examples. The refined text preserves discussions on port inventories and steel margins while placing more emphasis on subdued demand, thin profit margins, and bearish market sentiment as key indicator of prediction. Accordingly, the case-based explanations from the original text focus more on inventory management and short-term stable patterns, while those in the refined text highlight demand contraction, production constraints, and macroeconomic uncertainty, which is more consistent with a decreasing trend. Furthermore, the reasoning on time series provides a complementary view for predicting raw material price trends. The time series explanations identify declining price movements across multiple indices. In general, the multi-modal explanations based on matched prototypes from TimeXL demonstrate its effectiveness in capturing relevant raw material market condition for both predictive and explanatory analysis.

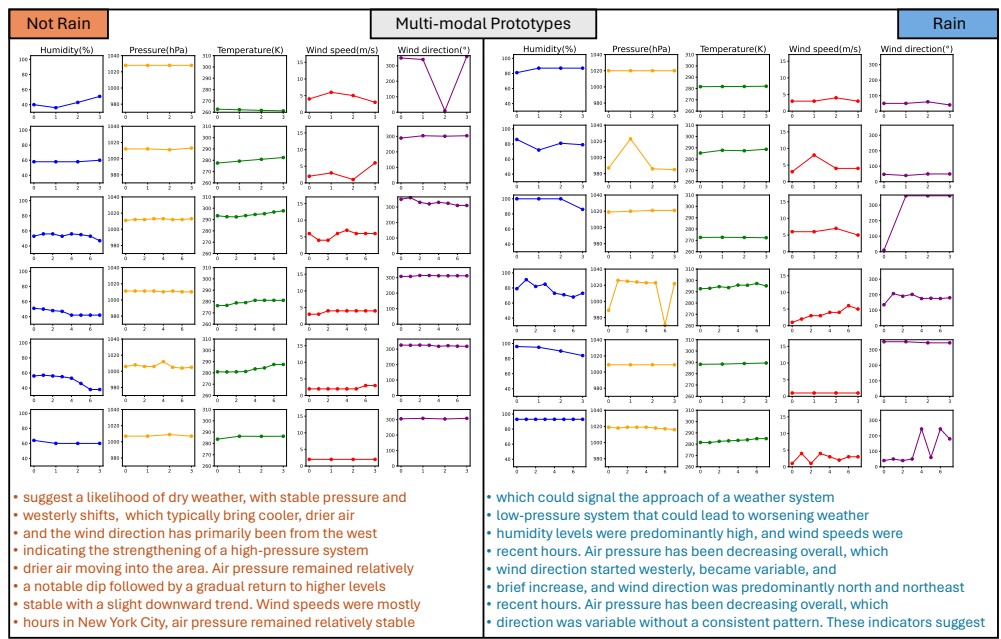

Figure 9: More multi-modal prototypes learned from the Weather dataset. Each row in the figure represents a time series prototype.

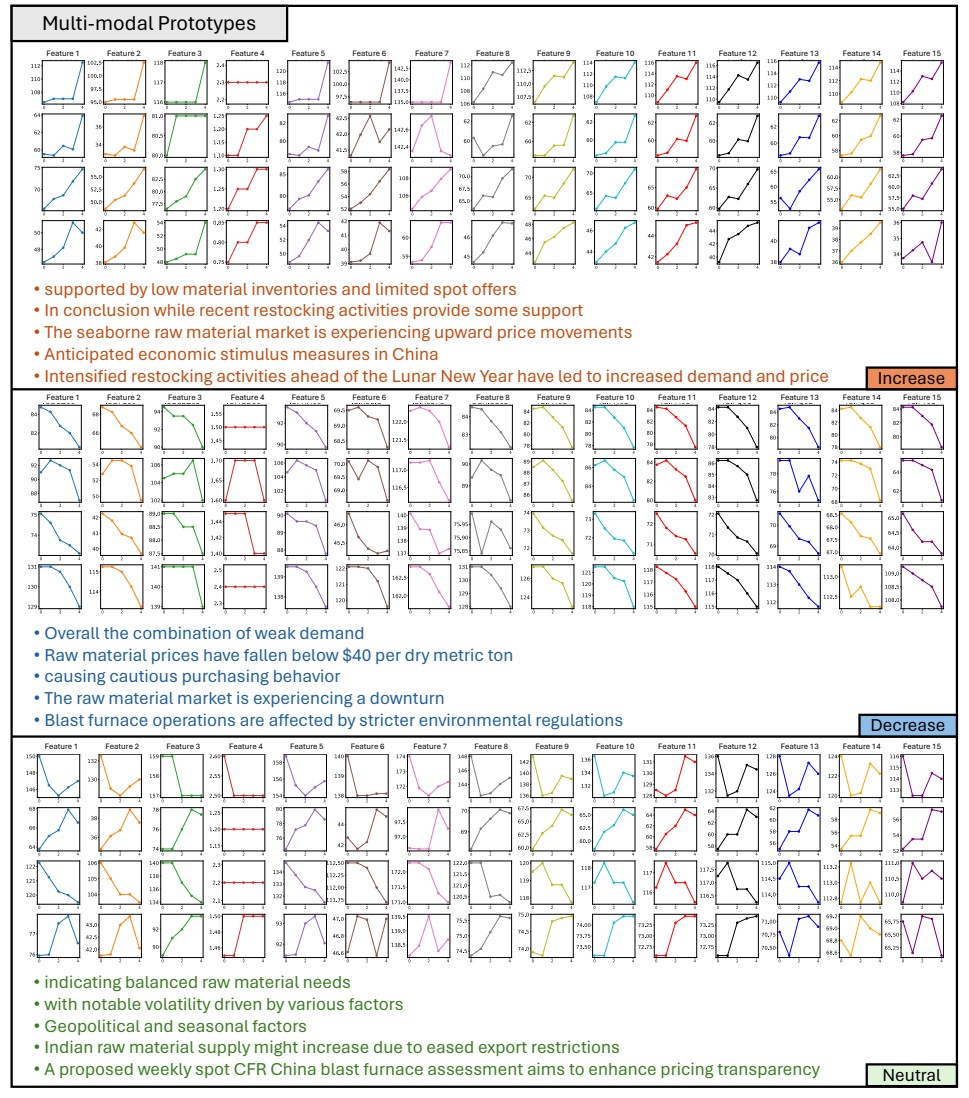

Figure 10: Key multi-modal prototypes learned from the Finance dataset. Each row in the figure represents a time series prototype.

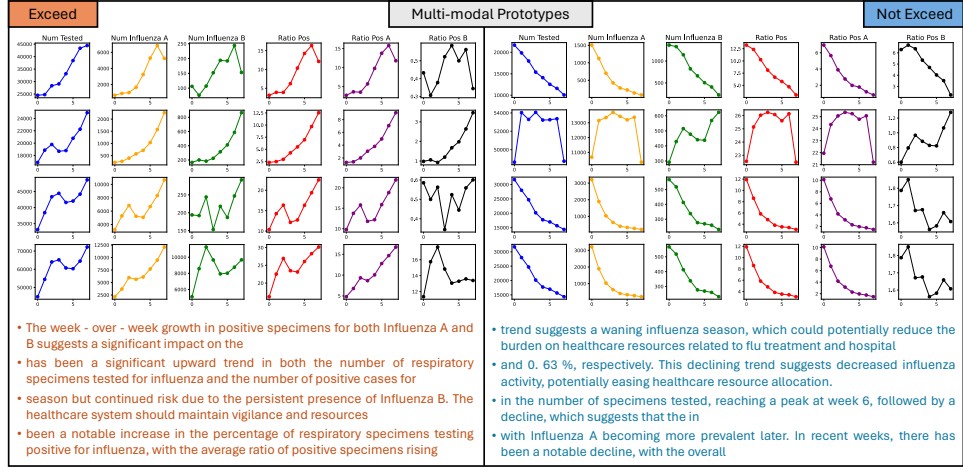

Figure 11: Key multi-modal prototypes learned from the Healthcare (Test-Positive) dataset. Each row in the figure represents a time series prototype.

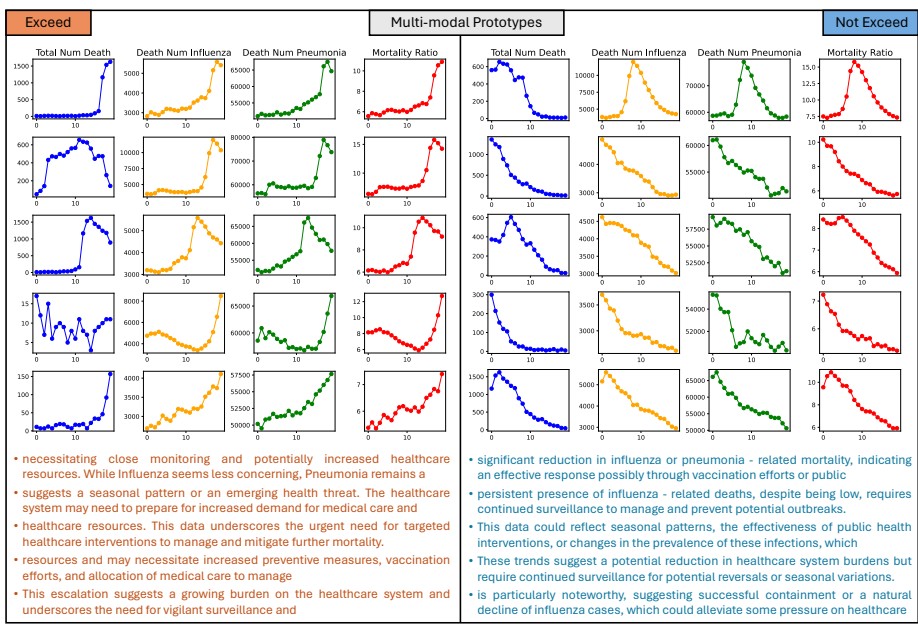

Figure 12: Key multi-modal prototypes learned from the Healthcare (Mortality) dataset. Each row in the figure represents a time series prototype.

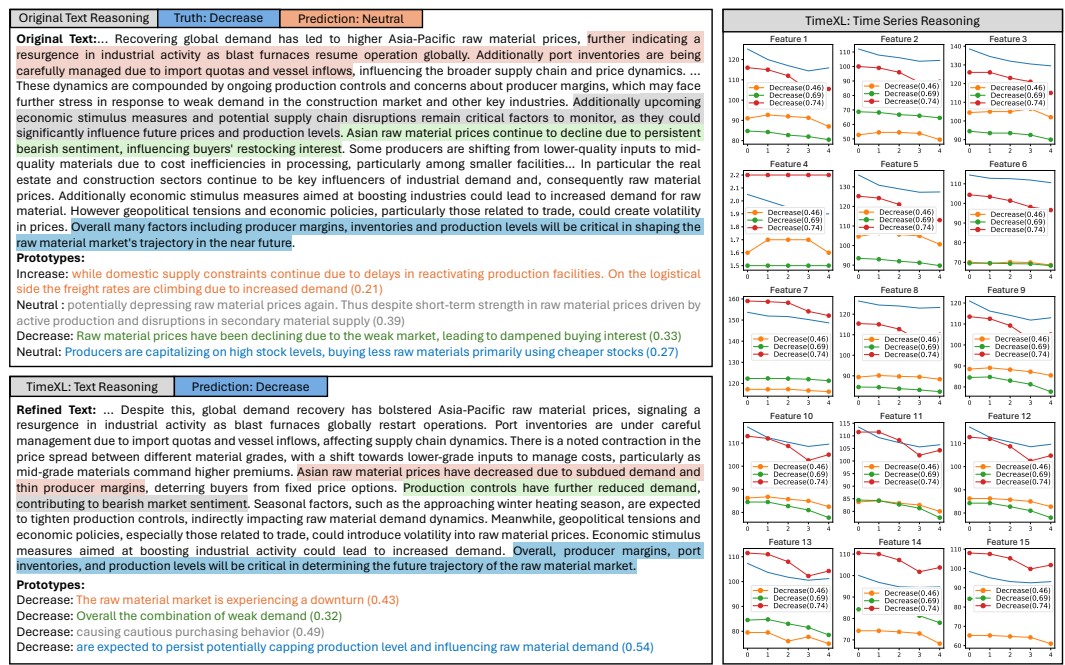

Figure 13: Multi-modal case-based reasoning example on the Finance dataset.

# E  Designed Prompts for Experiments

In this section, we provide our prompts for prediction LLM in Figure 14 (and a text-only variant for comparison, in Figure 15), reflection LLM in Figures 16, 17 18, as well as refinement LLM in Figure 19. Note that we adopt a generate-update-summarize strategy to effectively capture the reflective thoughts from training samples with class imbalances, which is more structured and scalable. We make the whole training texts into batches. First, the reflection LLM generates the initial reflection (Figure 16) by extracting key insights from class-specific summaries, highlighting text

patterns that contribute to correct and incorrect predictions. Next, it updates the reflection (Figure 17) by incorporating new training data, ensuring incremental and context-aware refinements. Finally, it summarizes multiple reflections from each class (in Figure 18) into a comprehensive guideline for downstream text refinement. This strategy consolidates knowledge from correct predictions while learning from incorrect ones, akin to the training process of deep models.

---

**System Prompt**

Your job is to act as [specific role]. You will be given a summary of [data description] and related prototypes that you can refer to. Based on this information, your task is to predict [task description].

**User Prompt**

Your task is to [task description]. First, review the following [number of prototypes] prototype text segments and outcomes, so that you can refer to when making predictions.

Prototype #1: [text prototype]
Corresponding Segment#1: [input text segment]
Relevance Score: [similarity score]
Outcome #1: [options]

...

Next, review the [situation] :
Summary: [text input]

Based on your understanding, predict the outcome of [situation]. Respond your prediction with [options]. Response should not include other terms.

---

Figure 14: Prompt for prediction LLM

---

**System Prompt**

Your job is to act as [specific role]. You will be given a summary of [data description]. Based on this information, your task is to predict [task description].

**User Prompt**

Your task is to [task description]. First, review the [situation] :

Summary: [text input]

Based on your understanding, predict the outcome of [situation]. Respond your prediction with [options]. Response should not include other terms.

---

Figure 15: Prompt for prediction with text only

---

**System Prompt**

You are an advanced reasoning agent that can improve the quality of [domain] summary based on self reflection. You will be given the summaries and [correct flag] predictions of [situation]. Your task is to learn some reflections that guides the refinement of [domain] summaries.

**User Prompt**

Your task is to analyze the provided [domain] summaries with [correct flag] predictions, in order to generate a reflection report improving its quality for [situation] prediction.

Review the following [number of summaries] [domain] summaries with [ground truth] actual outcomes and [prediction] predictions.

Summary #1: [text input]
Actual Outcome #1: [ground truth]
Prediction #1: [prediction]

...

Based on your analysis, write a high-quality reflection report that summarizes key phrases or sentences that led to correct predictions of [situation] / commonly misinterpreted and overlooked phrases or sentences that led to incorrect predictions of [situation].

Use precise terms to convey a clear and professional analysis, and avoid overly general statements. The report should be a comprehensive and informative paragraph, which can be generalized to refine similar [domain] summaries. Your response should not include other terms.

---

Figure 16: Prompt for reflection LLM: reflection generation

---

**System Prompt**

You are an advanced reasoning agent that can improve the quality of [domain] summary based on self reflection. You will receive a reflection report up to this point. You will also be given the summaries and [correct flag] predictions of [situation]. Your task is to learn some reflections and update the current report that guides the refinement of [domain] summaries.

---

**User Prompt**

Your task is to analyze the provided [domain] summaries with [correct flag] predictions, in order to update a reflection report improving its quality for [situation] prediction.

First, review the following reflection report up to this point: [current reflection report]

Next, review the following [number of summaries] [domain] summaries with [ground truth] actual outcomes and [prediction] predictions.

Summary #1: [text input]
Actual Outcome #1: [ground truth]
Prediction #1: [prediction]

...

Based on your analysis, write a high-quality reflection report that summarizes key phrases or sentences that led to correct predictions of [situation] / commonly misinterpreted and overlooked phrases or sentences that led to incorrect predictions of [situation].

Use precise terms to convey a clear and professional analysis, and avoid overly general statements. The report should contain incremental and context-aware updates, and can be generalized to refine similar [domain] summaries. Your response should not include other terms.

---

Figure 17: Prompt for reflection LLM: reflection update

---

**System Prompt**

You are an advanced summarization agent that can generate high-quality summarization. You will be given previously generated reflections for text refinement, from the correct and incorrect predictions of [domain] texts. Your current task is to summarize these long reflections to better guide financial text refinement.

---

**User Prompt**

Your task is to summarize the long reflections derived from previous predictions of [domain] contents. The goal is to generate a high-quality report aimed at improving the [domain] text quality for better predictive accuracy.

First, review the reflections from all combinations of possible predictions and actual outcomes: [reflection reports]

Based on your analysis, summarize the reflections of different scenarios and write a comprehensive report that provides guidelines to select the most important content in new [domain] texts where the actual outcome is unknown. Your response should keep the enough details, yet effective, to improve the text quality for downstream prediction. Your response should not include other terms.

---

Figure 18: Prompt for reflection LLM: reflection summarization

---

**System Prompt**

You are an advanced refinement agent designed to enhance the quality of [domain] summary. You will be provided with reflective thoughts analyzed from other summaries, and a summary that requires refinement. Your task is to generate a refined [domain] summary, by examining how reflective thoughts applied to the current summary.

---

**User Prompt**

Your task is to generate a refined weather summary from the current summary to improve its predictions of [situation]. First, review the following reflections that provide guidelines for refinement:

[final reflection report]

Next, review the current [domain] summary that describes [situation]:

Summary #1: [text input]

Based on your understanding, generate a new weather summary by selecting relevant content in the current summary, which provides insights crucial for understanding [situation]. Response should not include other terms.

---

Figure 19: Prompt for refinement LLM

# F Reflection Reports for Text Refinement

In this section, we provide the reflection reports after the first iteration, to demonstrate the reflective thoughts by accessing real-world contexts.

---

**Reflection Summary – Weather (New York)**

**Comprehensive Report on Refining Weather Predictions**
The review of weather summaries across various predictive scenarios reveals critical insights that can enhance the accuracy and reliability of weather forecasts. By integrating these refined elements, future weather texts can better anticipate actual outcomes. Below are the summarized guidelines for improving weather prediction texts:

**1.Key Indicators for Dry Conditions ("Not Rained" Predictions and Outcomes):**
1. **Stable and Slightly Increasing Air Pressure:** Consistent readings with minor upward trends suggest high-pressure systems, indicative of dry weather.
2. **Gentle to Moderate Wind Speeds:** Observations of stable wind speeds without gusts support non-precipitative forecasts.
3. **Variable Wind Directions (Northwesterly/Westerly):** Shifts from southwesterly to northwesterly/westerly directions bring cooler, drier air, reducing rain likelihood.
4. **Decreasing Daytime Humidity:** High early humidity followed by daytime decreases correlates with dry conditions.
5. **Typical Diurnal Temperature Patterns:** Normal temperature variations further support dry forecasts.
6. **Absence of Significant Weather Systems:** Lack of major air pressure or wind pattern changes reinforces stable, dry conditions.

**2.Indicators for Rainy Conditions ("Rained" Predictions and Outcomes):**
1. **Increasing Humidity Levels:** Significant humidity increases are strong rain indicators.
2. **Decreasing Air Pressure:** A downward pressure trend signals potential rain due to incoming low-pressure systems.
3. **Wind Conditions:**
   1. **Slight Wind Speed Increase:** Often precedes rain, particularly if observed later.
   2. **Easterly/Southeasterly Wind Directions:** Bring moisture-laden air, favoring rain.

**3.Common Misinterpretations ("Rained" Predictions with "Not Rained" Outcomes):**
1. **Pressure Stability Misinterpretation:** Minor pressure fluctuations without other indicators are weak rain predictors.
2. **Humidity and Fog Confusion:** Differentiate between humidity peaks indicating fog versus those suggesting rain.
3. **Overemphasis on Wind Changes:** Focus on sustained wind patterns rather than minor variations.
4. **Misjudged Temperature Fluctuations:** Evaluate temperature changes alongside corroborative indicators like pressure drops.
5. **Dynamic Weather System Misinterpretation:** Recognize stable atmospheric conditions to avoid false rain predictions.

**4.Overlooked Conditions ("Not Rained" Predictions with "Rained" Outcomes):**
1. **Humidity Peaks and Variability:** High peaks or fluctuations often precede rain; they shouldn't be underestimated.
2. **Wind Direction Shifts:** Changes to southerly or easterly directions can signal moisture influx.
3. **Temperature and Humidity Interactions:** Combined changes may offset stable pressure, indicating rain potential.
4. **Stable Pressure with Other Variables:** Consider pressure stability with high humidity and wind changes collectively.

By focusing on these detailed indicators and refining the interpretation of weather elements, future weather texts can improve prediction quality, providing more accurate and reliable forecasts.

Figure 20: Reflection summary for text refinement on the Weather dataset.

---

**Reflection Summary – Finance (Raw Material)**

**Comprehensive Guidelines for Enhancing Financial Text Quality in Predictive Analysis**
To improve the accuracy of financial predictions, it's crucial to refine the selection and emphasis of content in financial summaries. Based on reflections from past predictions, the following guidelines highlight the essential elements to consider for optimizing text quality:

**1.Key Indicators for Price Trends:**
1. For **increases**, focus on indicators such as robust demand from major consumers, strong profit margins in key manufacturing sectors, and preferences for high-grade raw materials. Also, consider tight supply chains, low inventories, and economic stimulus measures.
2. For **decreases**, emphasize rising inventory levels, weak demand, and operational adjustments. Bearish market sentiment and geopolitical tensions should be noted as well.
3. For **neutral** outcomes, identify market equilibrium indicators like balanced supply-demand dynamics, stable production operations, and moderate buyer behavior.

**2.Contextual Influences:**
1. Consider the impact of seasonal trends, restocking activities, and economic stimuli on demand fluctuations. Recognize how these factors may cause temporary market shifts rather than long-term trends.
2. Analyze global trade and supply chain disruptions to assess their potential to cause short-term volatility or stability rather than sustained changes.

**3.Regulatory and Policy Factors:**
1. Understand the implications of environmental regulations and geopolitical policies on supply and demand. These elements can significantly alter market dynamics and should be carefully integrated into analyses.

**4.Market Sentiment and Related Markets:**
1. Assess market sentiment and futures movements, recognizing that positive futures alignments often indicate underlying demand. Consider interconnected markets, such as related raw materials sectors , for their influence on overall demand.

**5.Strategic and Operational Adjustments:**
1. Pay attention to how producers and buyers adjust operations and procurement strategies in response to market conditions. These adjustments can affect demand patterns and price stability.

**6.Avoiding Common Misinterpretations:**
1. Avoid overemphasizing short-term trends as indicators of sustained market shifts. Be wary of speculative influences and temporary policy announcements that may not have immediate or lasting effects.
2. Recognize high inventory levels as potential strategic buffers and consider their role in stabilizing prices during off-peak periods.

By integrating these guidelines, financial summaries can provide a more comprehensive and balanced analysis, enhancing the predictive accuracy of unknown market outcomes. The focus should remain on understanding and articulating the complexities of market dynamics to better align predictions with actual trends.

Figure 21: Reflection summary for text refinement on the Finance dataset.

---

**Reflection Summary – Healthcare (Test-Positive)**

**Comprehensive Report on Improving Healthcare Text Quality for Predictive Accuracy**
**Introduction**
This report synthesizes reflections from past analyses of healthcare summaries concerning influenza positivity rates. The goal is to enhance the quality of these texts to improve predictive accuracy for future assessments where actual outcomes are unknown.

**Key Indicators for Accurate Predictions**
**1.Declining Trends for "Not Exceed" Predictions**
1. **Emphasis on Decline**: Correct predictions for outcomes that did not exceed the average often highlighted a clear decline in positivity rates. Terms such as "notable decline," "steady decrease," and "significant reduction" are crucial.
2. **Comparative Analysis**: A pronounced reduction in Influenza A and B, particularly Influenza A, supports accurate assessments.
3. **Testing Volume and Ratios**: A decrease in testing volume, coupled with positivity rates remaining below the average, strengthens predictions.
4. **Low Positivity Percentages**: Explicit references to recent positivity rates consistently falling below the long-term average are vital.

**2.Indicators of "Exceed" Predictions**
1. **Marked Increases**: Substantial increases in positivity rates, highlighted by specific numerical comparisons, indicate an "exceed" outcome.
2. **Influenza Strain Dominance**: Details on the predominance of Influenza A or B, with their impact on overall positivity rates, are significant.
3. **Rising Testing Volumes**: Increased specimen testing, peaking with positivity rates, suggests heightened incidence and surveillance.
4. **Healthcare System Impact**: References to increased hospitalizations and medical service demand provide context and validate exceeding predictions.
5. **Temporal Patterns**: Tracking weekly peaks aids in understanding and predicting heightened influenza activity.

**Common Misinterpretations**
•**Relative vs. Absolute Increases**: Misinterpretations often arise from conflating relative increases with exceeding the average positivity rate. It is essential to compare these increases explicitly with historical averages.
•**Contextual Misalignment**: Phrases highlighting rises in positivity rates without contextual alignment to the average threshold can mislead predictions.

**Guidelines for Text Refinement**
•**Integrate Clear Comparisons**: Ensure summaries incorporate direct comparisons to historical averages to avoid misjudgments.
•**Maintain Trend Clarity**: Differentiate clearly between increasing trends and those that actually surpass the average positivity rate.
•**Include Contextual Analysis**: Provide context for increases in positivity rates by detailing the impact on healthcare resources and system strain.
•**Highlight Temporal Data**: Incorporate temporal patterns and weekly peaks to aid in nuanced predictive analysis.

By adhering to these guidelines, healthcare texts can be refined to enhance predictive accuracy, ultimately aiding in proactive healthcare planning and resource allocation.

Figure 22: Reflection summary for text refinement on the Healthcare (Test-Positive) dataset.

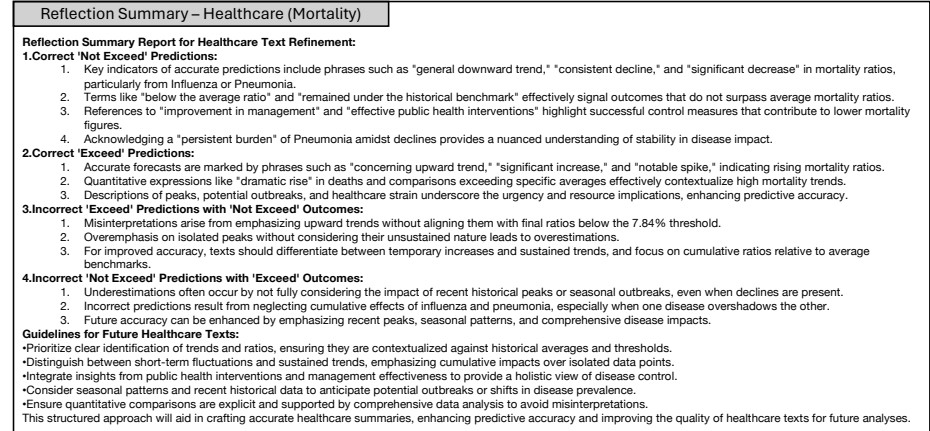

Figure 23: Reflection summary for text refinement on the Healthcare (Mortality) dataset.

# G TimeXL for Regression-based Prediction: A Finance Demonstration

In this section, we provide a demonstration of using TimeXL for regression tasks. Two minor modifications adapt TimeXL for continuous value prediction. First, we add a regression branch in the encoder design, as shown in Figure 24. On the top of time series prototype layers, we reversely ensemble each time series segment representation as a weighted sum of time series prototypes, and add a regression head (a fully connected layer with non-negative weights) for prediction. Accordingly, we add another regression loss term (*i,e.,* MSE and MAE) to the learning objectives. Second, we adjust the prompt for prediction LLM by adding time series inputs and requesting continuous forecast values, as shown in Figure 25. As such, TimeXL is equipped with regression capability.

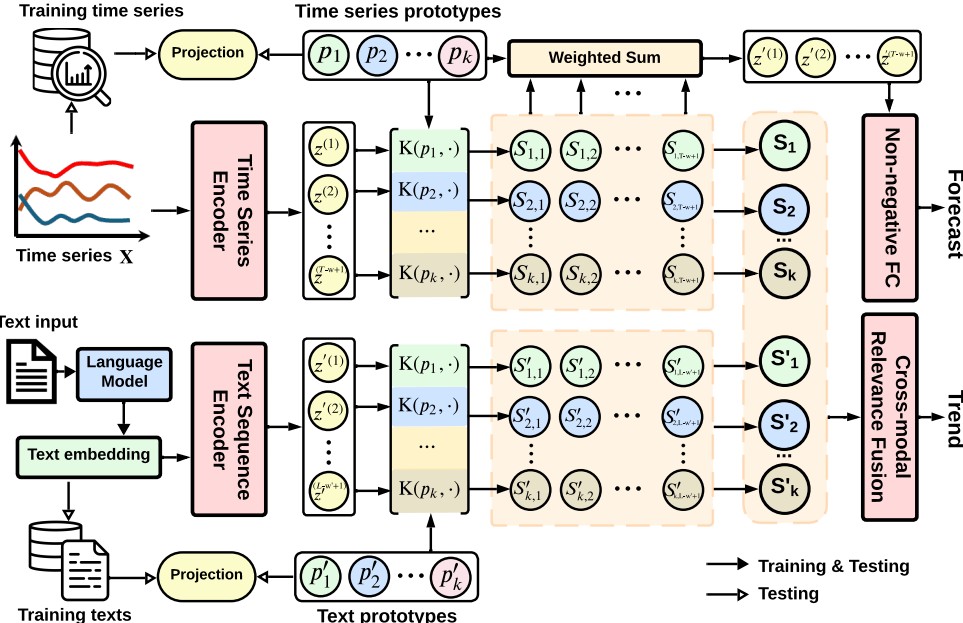

Figure 24: Multi-modal prototype-based encoder design in TimeXL for regression tasks.

```
┌─────────────────────────────────────────────────────────────────────────┐
│ ┌──────────────┐                                                          │
│ │ System Prompt │                                                         │
│ └──────────────┘                                                          │
│ Your job is to act as [specific role]. You will be given a summary of [data description] and related │
│ prototypes that you can refer to. Based on this information, your task is to predict [task description]. │
└─────────────────────────────────────────────────────────────────────────┘

┌─────────────────────────────────────────────────────────────────────────┐
│ ┌────────────┐                                                            │
│ │ User Prompt │                                                           │
│ └────────────┘                                                            │
│ Your task is to [task description]. First, review the following [number of prototypes] prototype text │
│ segments and outcomes, so that you can refer to when making predictions. │
│                                                                           │
│ Prototype #1: [text prototype]                                            │
│ Corresponding Segment#1: [input text segment]                             │
│ Relevance Score: [similarity score]                                       │
│ Outcome #1: [options]                                                     │
│                                                                           │
│                                    ...                                    │
│                                                                           │
│ Next, review the [situation] :                                            │
│ Summary: [text input]                                                     │
│                                                                           │
│ Finally, review the [domain] record of [situation] : [time series values] │
│                                                                           │
│ Based on your understanding, predict the outcome of [situation], followed by the value of [domain] │
│ record. Respond your prediction with [options] and [numerical value]. Response should not include │
│ other terms.                                                              │
└─────────────────────────────────────────────────────────────────────────┘
```

Figure 25: Prompt for prediction LLM in regression tasks.

Table 8: Performance evaluation for regression tasks.

| Model | RMSE | MAE | MAPE(%) |
|---|---|---|---|
| **DLinear [75]** | 7.871 | 6.400 | 4.727 |
| **Autoformer [76]** | 7.215 | 5.680 | 4.263 |
| **Crossformer [77]** | 7.205 | 5.313 | 3.808 |
| **TimesNet [78]** | 6.978 | 4.928 | 3.512 |
| **iTransformer [79]** | 5.877 | 4.023 | 2.863 |
| **TSMixer [80]** | 7.447 | 5.509 | 3.911 |
| **TimeMixer [81]** | 6.651 | 4.703 | 3.349 |
| **FreTS [82]** | 7.098 | 4.886 | 3.460 |
| **PatchTST [5]** | 5.676 | 4.042 | 2.853 |
| **LLMTime [83]** | 11.545 | 5.300 | 3.774 |
| **PromptCast [63]** | 4.728 | 3.227 | 2.306 |
| **OFA [53]** | 6.906 | 4.862 | 3.463 |
| **Time-LLM [56]** | 6.396 | 4.534 | 3.238 |
| **TimeCMA [57]** | 7.187 | 5.083 | 3.620 |
| **FSCA [58]** | 5.511 | 3.873 | 2.720 |
| **MM-iTransformer [35]** | 5.454 | 3.789 | 2.687 |
| **MM-PatchTST [35]** | 5.117 | 3.493 | 2.491 |
| **TimeCAP [65]** | 4.456 | 3.088 | 2.196 |
| TimeXL | **4.161** | **2.844** | **2.035** |

We conduct the experiments on the same Finance dataset, where the target value is the raw material stock price. The performance of all baselines and TimeXL is presented in Table 8. Consistent with the classification setting, TimeXL achieves the best results, and the multi-modal variants of state-of-the-art baselines (MM-iTransformer and MM-PatchTST) benefit from incorporating additional text data. This observation is further supported by the ablation study in Table 9. We also note that the text modality offers complementary information for LLMs, enhancing numerical price predictions. Furthermore, the identified prototypes provide contextual guidance, leading to additional performance gains. Overall, TimeXL outperforms all variants, underscoring the effectiveness of mutually augmented prediction. We also provide an iteration analysis to show the effectiveness of reflective and refinement process, as shown in Table 10. The prediction performance quickly improves and stabilizes over iterations, which underscores the alternation steps between predictions and reflective refinements.

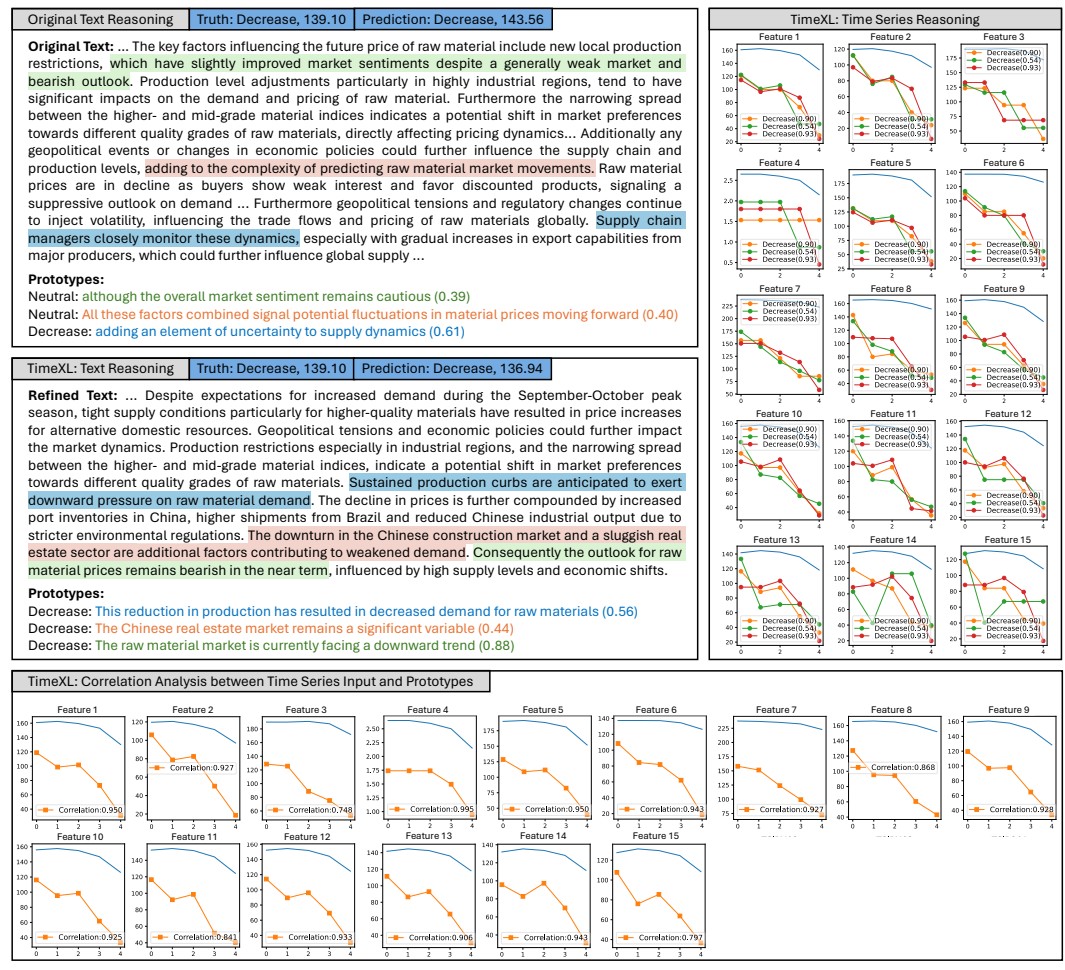

Figure 26: Multi-modal case-based reasoning example on the Finance dataset for regression tasks.

Table 9: Ablation studies of TimeXL for regression tasks.

| Variants | RMSE | MAE | MAPE(%) |
|---|---|---|---|
| Time Series Prototype-based Encoder | 4.287 | 3.001 | 2.140 |
| Multi-modal Prototype-based Encoder | 4.198 | 2.891 | 2.064 |
| Prediction LLM - Time Series + Text | 4.600 | 3.121 | 2.226 |
| Prediction LLM - Time Series + Text + Prototype | 4.352 | 3.003 | 2.165 |
| TimeXL | **4.161** | **2.844** | **2.035** |

Table 10: Iterative analysis of TimeXL for regression tasks.

| Iteration | RMSE | MAE | MAPE(%) |
|---|---|---|---|
| Original | 4.344 | 2.951 | 2.103 |
| 1 | 4.224 | 2.883 | 2.069 |
| 2 | 4.161 | 2.844 | 2.035 |
| 3 | 4.174 | 2.849 | 2.036 |

Finally, we provide a case study on the test set of the Finance dataset that demonstrates the effectiveness of TimeXL's reasoning process for regression tasks, as shown in Figure 26. We present

the top three most similar prototypes allocated for both time series and text modalities, where we have two key observations. 1) Even if the trend predictions align correctly in both original and refined texts, the prototype allocation varies regarding market sentiments. In the original texts, prototypes from the neutral class are also allocated, highlighting a mixed sentiment and potential fluctuations, consequently deteriorating numerical prediction accuracy from both the encoder and the LLM. In the refined texts, prototypes instead capture the underlying bearish market sentiment for raw materials, emphasizing decreased demand and weakened market conditions, effectively aligning bearish market sentiment with numerical predictions. 2) The allocated top similar time series prototypes also exhibit highly similar trends compared to the original input series, offering robust reference points for numerical prediction. To quantitatively assess it, we aggregate the retrieved prototypes using their similarity scores and compute the correlation between the input time series and the aggregated prototype series. The observed strong correlations underscores TimeXL's efficacy in capturing intrinsic temporal patterns and providing predictive insights into future numerical stock values based on similar historical patterns.

## H    More Detailed Discussions of Related Work

Due to space limitations, we provide additional details on related methods here, expanding on the overview in the main text.

### H.1    Multi-modal Time Series Analysis.

In recent years, multi-modal time series analysis has gained significant traction in diverse domains such as finance, healthcare and environmental sciences [28, 11, 13]. Multiple approaches have been proposed to model interactions across different modalities, such as multi-modal fusion and cross-modal alignment and so on. Multi-modal fusion captures complementary information to enhance time series modeling, which are commonly implemented via addition [35] and concatenation [29] at the multiple levels. For example, Liu *et al.* [35] fuse predictions from both time series and text modalities via learnable linear weighted addition, further enhancing the predictive performance of state-of-the-art forecasters. In parallel, cross-modal alignment aims to capture the association between time series and other modalities for downstream tasks, where common techniques include attention mechanism [12, 30, 86, 31], gating functions [32], and contrastive learning [33, 34]. For instance, Bamford *et al.* [14] align time series and text in a shared latent space using deep encoders, enabling retrieval of specific time series sequences based on textual queries. In addition, Zheng *et al.* [33] perform causal structure learning in multi-modal time series by separating modality-invariant and modality-specific components through contrastive learning, enabling root cause analysis. Besides method developments, recent research efforts have also advanced the field through comprehensive benchmarks and studies [35, 36, 37], highlighting the incorporation of new modalities to boost task performance. Meanwhile, others explore representing time series as other modalities (*e.g.*, image, text, table, graph) [38, 39, 40, 41, 42] or generating synthetic modalities [32, 43] to facilitate downstream tasks. Nevertheless, these studies tend to focus primarily on improving numerical accuracy. The deeper and tractable rationale behind *why* or *how* the textual or other contextual signals influence time series outcomes remains underexplored.

### H.2    Time Series Explanation

Recent studies have explored diverse paradigms for time series interpretability. Gradient-based and perturbation-based explanations leverage saliency methods to highlight important features at different time steps [44, 45], where some methods explicitly incorporate temporal structures into models and objectives [46, 47]. Meanwhile, surrogate approaches also offer global or local explanations by evaluating the importance of time series features with respect to prediction. These methods include the the generalization of Shapley value to time series [48], the formulation of model behavior consistency via self-supervised learning [49], and the usage of information-theoretic guided objectives for coherent explanations [50]. In contrast to saliency or surrogate-based explanations, we adopt a *case-based reasoning* paradigm [25, 26, 27], which end-to-end generates predictions and built-in explanations from learned prototypes. Our work extends this approach to multi-modal time series by producing human-readable reasoning artifacts for both the temporal and contextual modalities.

### H.3 LLMs for Time Series Analysis

The rapid development of LLMs [15, 16] has begun to inspire new directions in time series research [51, 52]. Many existing methods fine-tune pre-trained LLMs on time series tasks and achieving state-of-the-art results in forecasting, classification, and beyond [53, 54, 55]. The textual data (*e.g.*, domain instructions, metadata, and dataset summaries) are often encoded as prefix embeddings to enrich time series representations [56, 57, 58, 59]. These techniques also contribute to the emergence of time series foundation models [55, 60, 61, 62]. An alternative line of research leverages the zero-shot or few-shot capabilities of LLMs by directly prompting pre-trained language models with text-converted time series [63] or context-laden prompts representing domain knowledge [24], often yielding surprisingly strong performance in real-world scenarios. Furthermore, LLMs can act as knowledge inference modules, synthesizing high-level patterns or explanations that augment standard time series pipelines [64, 65, 66]. For instance, Wang *et al.* [66] construct a large news database for method development and perform text-to-text prediction by fine-tuning an LLM with dynamically selected news, which uses reflective selection logic to incorporate matched social events and thus augment numerical prediction. Building on this trend, recent works have explored time series reasoning with LLMs by framing tasks as natural-language problems [68], including time series understanding & question answering (*e.g.*, trend and seasonal pattern recognition, similarity comparison, causality analysis, *etc.*) [67, 69, 70, 71, 72], as well as language-guided inference for standard zero-shot time series tasks [67, 69]. Compared to existing LLM-based methods, our approach provides a unique synergy between an explainable model and LLMs through iterative, grounded, and reflective interaction. It explicitly provides case-based explanations from both modalities, highlighting key time series patterns and refined textual segments that contribute to the prediction.

## I Limitations

Our main focus has been on classification-based time series prediction to emphasize the explanation capabilities in real-world multi-modal applications, while we also provide a detailed regression-based prediction setting in the appendix. Extending our framework to broader regression-based tasks under the multi-modal contexts, such as long-term time series forecasting, and time series forecasting with domain shifts, remains an important direction for us to explore in the future work.

## J Broader Impacts

Our paper presents advancements in explainable multi-modal time series prediction by integrating time series encoders with large language model-based agents. The broader impact of this work is multifaceted. It has the potential to support high-stakes decision-making in domains such as finance and healthcare by delivering more accurate predictions accompanied by reliable case-based explanations that lead to more robust analyses. The social impact of this work stems from its potential to provide a new paradigm for analyzing real-world multi-modal time series through the integration of emerging AI tools such as language agents. We emphasize the importance of responsible and ethical deployment of LLMs in time series analysis to ensure that such advancements lead to positive societal outcomes.

