# OpenReview forum: "TimeXL: Explainable Multi-modal Time Series Prediction with LLM-in-the-Loop"
_NeurIPS.cc/2025/Conference — NeurIPS 2025 poster_

### Official Review · Reviewer_vdBq · 2025-06-30

**Clarity:** 3
**Significance:** 3
**Originality:** 3
**Rating:** 4
**Confidence:** 4

**Summary:**

1. A prototype-based encoder producing transparent, case-based rationales.

2. Iterative refinement via LLM-driven reasoning to improve predictions and text quality.

3. Empirical validation showing superior performance over baselines.

**Questions:**

1. Generalization to Non-Textual Modalities: The framework relies heavily on textual context. How might it perform with other auxiliary modalities (e.g., images, sensor data)?

2. Scalability: The iterative refinement process may incur computational costs. Are there optimizations for large-scale deployment?

3. Ethical Risks: The reflection LLM might amplify biases in textual data. How is bias mitigation handled?

**Ethical Concerns:**

["NO or VERY MINOR ethics concerns only"]

**Final Justification:**

Ethical Risks is unresolved. As the authors explain, "Adapting such evaluations to our setting would require additional effort, such as task-specific annotations."

I keep my score.

**Limitations:**

The authors acknowledge limitations (e.g., focus on classification, future extension to regression).

Suggestions:

Discuss computational trade-offs of the refinement loop.

Expand on ethical safeguards for LLM-generated text (e.g., bias audits).

**Quality:**

3

**Strengths And Weaknesses:**

Strengths:

1. The methodology is technically sound, with rigorous experiments across diverse datasets (weather, finance, healthcare). The ablation studies validate design choices (e.g., multi-modal fusion, LLM integration).

2. Well-organized with clear figures (e.g., workflow). The appendix supplements key details (prompts, reflection reports).

3. Addresses a critical gap in explainable AI for time series, with potential high-stakes applications (e.g., healthcare, finance). The 8.9% AUC improvement is notable.

4. Novel integration of prototype-based encoders with LLMs for iterative refinement. The closed-loop workflow is innovative.

Weaknesses:

1.  The regression task demonstration (Appendix F) is less detailed compared to classification. Broader regression applications (e.g., long-term forecasting) are noted as future work.

2. Some technical terms (e.g., "cross-modal relevance fusion") could be further clarified for non-experts.

3. While the framework is general, domain-specific limitations (e.g., reliance on textual context quality) could be explored further.

4. Similar LLM-augmented approaches exist (e.g., TimeCAP), though TimeXL’s focus on interpretability is distinctive.

---

> ### Author Rebuttal · Authors · 2025-07-31
>
> We sincerely appreciate the reviewer’s constructive feedback and address the concerns below.
>
> **W1 Details of regression tasks**
>
> We thank the reviewer for the helpful feedback. While we acknowledge that our primary emphasis is on classification settings, we would like to clarify that the regression task in Appendix F (Pages 26-29) is methodologically complete and systematically evaluated. Appendix F includes a modified methodology (Figure 24) and prompt design (Figure 25) tailored for regression tasks, along with performance evaluation against state-of-the-art forecasting baselines (Table 6) and regression visualizations (Figure 26). We also provide supporting analyses including an ablation study (Table 7), iterative refinement analysis (Table 8), and a case study (Figure 27). That said, we appreciate the suggestion and agree that broader regression scenarios, such as long-horizon forecasting, deserve further investigation. We will also consider expanding the regression coverage in the main text or supplementing further details in an updated version of the manuscript.
>
> **W2 Clarification of terms**
>
> Thanks for the helpful suggestion. As noted in lines 162-163, we consider the cross-modal relevance by computing similarity scores between prototypes and input segments across both modalities, which are then fused using a non-negative weight matrix to obtain prediction probabilities. We will revise the text and add more details to make this term clearer for non-expert readers, and pay closer attention to other terms.
>
> **W3 Exploration of domain-specific limitations such as reliance on textual context quality**
>
> We sincerely thank the reviewer for pointing this challenge out. Our method is able to handle noisy or redundant textual inputs by using iterative and reflective feedback from training data to improve text quality for more informative prediction. This makes the framework robust to moderate imperfections in the textual context.
>
> However, we agree with the reviewer that in domains where textual context is extremely sparse, unstructured or uninformative, the effectiveness of our method could be limited. In such cases, additional strategies from domain-experts such as designing domain-specific prompts or methodology developments such as cross-modal augmentation and external knowledge-base retrieval would be needed to address this challenge. We will view this as an important direction for future work and discuss them in our manuscripts.
>
> **W4 TimeXL’s advantage over existing LLM-augmented approaches**
>
> Thanks for acknowledging our contribution in terms of interpretation! We would like to slightly justify that, besides the built-in interpretation mechanism and performance improvement, our approach provides a unique synergy and mutual augmentation between a multi-modal time series model and LLMs through iterative, grounded, and reflective interaction.
>
> **Q1 Generalizing TimeXL to non-textual modalities**
>
> Thanks for this question! Our proposed method has the capability to handle other auxiliary modalities for interpretable prediction, where the key idea is to use a prospective modality encoder with a prototype design. For example, to support an image modality, we can use a 2D convolutional encoder to extract spatial features from local regions. Class prototypes can then be defined over the resulting representations, with similarity scores computed via Equation 2 and fused with other modality scores through Equation 3 for outcome prediction. After training, these interpretable prototypes can be identified as representative image regions via Equation 6 (e.g., local abnormal regions of an X-ray in medical imaging). These prototypes enable case-based visual explanations during test inference, when jointly analyzed with other modalities (e.g., lab-test time series, clinical notes, which our framework currently supports).
>
> **Q2 and L1 Scalability and optimizations for large-scale deployment with trade-off**
>
> We thank the reviewer for pointing this practical question out and provide our discussion as follows. First, we would like to quantitatively discuss the computational cost of the iterative process regarding token usage and time. Following the prompts and strategy detailed in Appendix D (Pages 22-24), each iteration over  4000 training and validation samples (average length of 400 tokens or ~300 words) requires ~9.74 M input tokens and yield ~1.64 M output tokens. With GPT‑4o (with input latency as TTFT ~0.45 s and output speed ~156 tokens/s), it corresponds to ~3.9 hours serially per iteration (prediction takes ~0.5 h, reflection ~0.08 h, and refinement ~3.35 h). As prediction and refinement steps can also be parallelized, we can reduce the iteration time to ~12 min with 20 concurrent calls. Using a more efficient LLM base model Gemini-Flash-2.0 (evaluated in Appendix B, Pages 18-19) with TTFT ~0.3 s and ~236 output tokens/s, the iteration time can be further reduced to ~8 min.
>
> Note that we do not run reflection in the testing/deployment stage, but instead apply the fixed reflection selected during validation to refine the test texts (as noted in lines 245-250), reducing inference cost and latency. Based on our prompt lengths, the refinement and prediction steps take ~3.46 s per sample using GPT‑4o, and ~2.29 s per sample for Gemini-2.0-Flash, which is reasonable for deployment in real-world applications .
>
> Besides using more efficient LLM backbones and parallelized query as aforementioned, there are a few more strategies to optimize the computational cost for large-scale deployment. Firstly, we can control the number of iterations based on the trade-off between performance and cost. As shown in Figure 5 and discussed in Section 4.5, 1-2 iterations are often sufficient to see clear performance improvements. We can monitor performance gains on validation data after over iterations and terminate the loop early if the improvement falls below a predefined threshold. We can also do selective refinement by skipping training examples from the large scale data, which were already predicted accurately in previous iterations. These strategies help make our method more computationally tractable. We will include a detailed discussion in our updated version.
>
> **Q3 and L3 Ethical risk mitigation for LLM-generated texts**
>
> We appreciate the reviewer’s thoughtful comments regarding the bias in LLM generated texts. As we noted in the checklist, we tried to review and monitor the LLMs responses, using structured prompt templates (provided in Appendix D, Pages 22-24 and supplementary code), and the datasets are from the general domains, which are fairly neutral and not identity-sensitive. Nevertheless, we agree with the reviewer that this is an important concern and we will consider more formal assessments and countermeasures in future work, particularly when applying the framework to sensitive datasets such as clinical data.

---

> > ### Comment · Reviewer_vdBq · 2025-08-05
> > **Thank you for your reply**
> >
> > Thank you for your answer, which resolved my concerns.
> >
> > I'll keep my original score.

---

> > ### Comment · Reviewer_Vxdr · 2025-08-06
> >
> > I agree with reviewer vdBq's concern about the reflection LLM introducing further bias or inaccuracies that in more critical settings could lead to further risks. This is something I was trying to get at with my own Q3. Would benchmarking the LLM pipeline results against factuality tests like TruthfulQA or HallusionBench help verify that there is not significant drift in the underlying information being passed on?

---

> > > ### Comment · Reviewer_vdBq · 2025-08-06
> > > **comments on L3 of Vxdr**
> > >
> > > Thank you (Vxdr) for raising the question L3: I believe the authors are unable to measure how much information in the test set is already known by the LLM in advance.

---

> > > > ### Author Response · Authors · 2025-08-08
> > > >
> > > > We sincerely appreciate your engagement in the discussion. As you pointed out, assessing potential overlap between benchmark datasets and LLM pretraining corpora remains an open and shared challenge in the field. Here we would like to take this opportunity to provide preliminary assessments and briefly elaborate on potential countermeasures to help mitigate related risks.
> > > >
> > > > We first adopt a multiple-choice recall probe as a proxy to assess whether the model memorizes test samples (i.e., potential inclusion in the pretraining corpora, or data contamination), similar to the data contamination quiz framework [1]. In this probe, the model is presented with k candidate lines (one from the held-out test set and k-1 distractors from other test items) without any contexts, and is asked to identify the correct one. If the model lacks memorization, performance should approximate random guessing (chance $\approx$ 1/k).
> > > >
> > > > We ran the probe on the testing data of the weather dataset, using gpt-4o-2024-08-06 with k=5 (chance = 1/5 = 20%) , and 200 questions for multiple-choice recall test. The accuracy is close to chance (23.0%, p value = 0.29 >> 0.05, binomial test) suggesting no statistically significant evidence of memorization of the test content by the LLM.
> > > >
> > > > As a complementary check, we also consider a prefix recall probe to assess potential memorization in a generative completion setting, following [2] and [3]. In the test, the model is given only with the initial p% of a held-out test sample (prefix) and is prompted to continue the text exactly as in the original sample. We evaluate two metrics: (1) Exact match - the generated continuation matches the reference suffix exactly, with a minimum match length threshold. (2) Near match - the generated continuation shares a longest common subsequence with the reference suffix whose length ratio is at least $\tau$ .
> > > >
> > > > We ran this probe on the testing data of weather using the same LLM with p% $\in$ {10%, 20%, 50%}, maximum continuation length = 128 tokens, and near-match threshold $\tau$ $\in$ {0.5, 0.7, 0.9}. For each prefix length, we sampled 200 test items. Even under a very lenient setting ($\tau$=0.5, exact-match length = 10 characters), results showed no exact matches and a maximum near-match rate of 0.04, suggesting no clear verbatim memorization in this generative-completion setting.
> > > >
> > > > We next discuss other methods that could help mitigate this issue. As a preventive measure, a good practice is to evaluate on datasets collected after the LLM’s pretraining cut-off date, when feasible, to help reduce contamination risk. One can also analyze the LLM’s output distributions to identify potential data contamination. The intuition is that if the LLM consistently generates identical or highly similar outputs across multiple samples, it is likely that the input was seen during pretraining [4]. A similar strategy can be applied at the token level for LLMs that support output token distribution [5]. We will include a discussion of this important challenge in our updated manuscript.
> > > >
> > > > [1] Data Contamination Quiz: A Tool to Detect and Estimate Contamination in Large Language Models
> > > >
> > > > [2] Quantifying Memorization Across Neural Language Models
> > > >
> > > > [3] A Comprehensive Analysis of Memorization in Large Language Models
> > > >
> > > > [4] Generalization or Memorization: Data Contamination and Trustworthy Evaluation for Large Language Models
> > > >
> > > > [5] Detecting Pretraining Data from Large Language Models

---

> > > > > ### Comment · Reviewer_vdBq · 2025-08-08
> > > > > **Thank you**
> > > > >
> > > > > Thank you for your detailed response to my concern.
> > > > >
> > > > > I have no other questions.

---

> > > ### Author Response · Authors · 2025-08-07
> > >
> > > We appreciate the reviewer’s thoughtful follow-up and engagement with reviewer vdBq’s comments.  We also thank the reviewer for bringing factuality benchmarks such as *TruthfulQA* and *HallusionBench* into our discussion. We agree that assessing factual consistency is an important direction to explore, particularly for detecting drift. Adapting such evaluations to our setting would require additional effort, such as task-specific annotations, but we view this as a promising area for the future work of multi-modal time series analysis. To help mitigate these risks, hallucination detection methods can be explored. A possible solution is to decompose the detection process into multiple steps, such as using a chain-of-verification strategy [1], where LLM first drafts an initial reflection, then generates verification questions, and independently answers them to check inconsistencies. We can also generate multiple reflection reports and assess the consistency between the original response and these generated ones [2]. The detection could also be made at different granularities, such as sentence-level and passage level, depending on the specific need and application context [3]. For example, sentence-level assessment may be preferred in critical scenarios. We will include a discussion of the aforementioned benchmarks and methods in the revised manuscript.
> > >
> > > Again, we sincerely appreciate the reviewer’s insights, which adds a valuable perspective for improving our framework’s reliability in future extensions.
> > >
> > > [1] Chain-of-Verification Reduces Hallucination in Large Language Models
> > >
> > > [2] SelfCheckGPT: Zero-Resource Black-Box Hallucination Detection for Generative Large Language Models
> > >
> > > [3] A Survey on Hallucination in Large Language Models: Principles, Taxonomy, Challenges, and Open Questions

---

### Official Review · Reviewer_Vxdr · 2025-07-02

**Clarity:** 2
**Significance:** 2
**Originality:** 3
**Rating:** 4
**Confidence:** 4

**Summary:**

The authors provide a new framework (TimeXL) for text+time series multimodal predictions using a series of prompts and a prototype-based encoder that generates more accurate predictions and contains a higher level of interpretability than previous approaches. The predict-critique-refine cycle applied to the encoder shows improved performance on select multimodal time-series classification tasks.

**Questions:**

Q1. Channel-based clustering seems fairly effective [link] for long-context time series predictions. I wonder if that might serve as a replacement for the prototype method instead? Have the authors considered other alternatives?

Q2. It seems that the prototypes are found based on the segment-level representations which seems like it may lead to the loss of long-term or irregular sequence information. Is there any way to verify this through longer-sequence benchmarks (even on non-multimodal methods since the prototypes are applied to the time series data as well)?

Q3. From the examples in figure 4, I’m curious what additional information seems to be added in either the original or refined input text. The same segments with matching prototypes in the original text “gradual increase in temperature” can also be found in the refined text, and moreover should be learned by the time-series model in the underlying sequence data anyway. It’s unclear what exactly is contributing to the higher performance. Perhaps investigating some of the classifications that changed to become true positives after introduction of the new framework would shed some light.

**Ethical Concerns:**

["NO or VERY MINOR ethics concerns only"]

**Final Justification:**

I find the approach novel and interesting, however I do think the final results could be strengthened and there are some questions and weaknesses which haven't been fully addressed by the authors. In particular the two remaining areas that make me hesitate to raise my rating:
1. The claims around explainability which are not well empirically grounded
2. The concerns raised by reviewer vdBq around potential safety / ethical risks related to concept drift or the introduction of further bias through the LLM approach.

**Limitations:**

L1. The increased performance brought by this method is reliant on the nature of the data in the text modality and its pairing with the hand-engineered prompts and LLM models selected. This seems somewhat brittle, though there are obvious advantages in performance that the authors have shown for some tasks. Improved methods for evaluating the quality of the text data would help practitioners understand when it is feasible or desired to apply a multimodal method like this one vs just using the time series data.

L2. The task generalization is evaluated in a limited fashion here and it may be the case that the classification tasks are particularly well-suited to this kind of fusion and iterative approach.

L3. It is not clear to me if there is an easy way of knowing how much of the current evaluation benchmarks may have leaked into the LLM pre-training data and whether the introduction of these models may be simply providing roundabout (though maybe not direct) access to the ground truth of the task.

**Quality:**

3

**Strengths And Weaknesses:**

S1. Good Performance. Clear performance gains on the Weather and Healthcare datasets do validate the hypothesis that introducing the prototype-encoder and refinement loop improves on previous multimodal methods.

S2. Interpretability through Prototypes. In general the prototype method is a helpful one for understanding similar patterns in data that may be related to a given inference. Other methods have shown the introduction of prototypes can improve performance on other modalities like vision/text, so it makes sense to try it here. It also acts as a helpful gut-check on the model’s reasoning. There is likely further room to develop these ideas in a more robust way.

W1. Interpretability Evals. The claim that “This closed-loop workflow… continuously boosts the framework’s performance and interpretability” is not empirically verified. While performance increases are reported and average +1.9% AUC, the authors claim that TimeXL achieves up to 8.9% improvement in AUC but it seems that the largest increase is 6.6% on the weather task (vs TimeCAP) while the other healthcare baselines seem to be already saturated. It may be useful to create or introduce other multimodal benchmarks that can highlight if there is significant differentiation in the method. For interpretability it seems some predictions are inspected and verified by the authors, which does looks decent, but is not a very robust evaluation strategy. Have the authors considered having subject-matter experts for each task-type evaluate the results and report on whether the interpretations make sense?

W2. Cost. There is no accounting of the latency or cost of either the training or inference method. TimeXL trains the encoder, calls a prediction, reflection and refinement LLM for every optimization step, then re-trains the encoder with the updated text. I imagine this significantly increases both training cost and inference latency compared to the encoder-only baseline. It would be helpful to articulate these costs so the right tradeoff between performance and cost can be evaluated. On many tasks its unclear that the additional interpretability and marginal performance gains would be worth the added complexity.

W3. Task Expansion. Expanding to other tasks is mentioned in the paper but based on the appendix evaluations on regression it seems there is a lot less performance to be gained in these other task areas. There are other multimodal time series datasets the authors can try evaluating on (e.g. TimeMMD) and this might give more insight into the task generalization of the method.

W4. Parameter Selection. Hand-tuned hyper parameters may not generalize well to other tasks. TimeXL’s final prediction is a linear combination of encoder and LLM outputs controlled by a weight, a, selected from validation data. Similar manual tuning applies to three prototype-regularisation coefficients and the number of prototypes injected into the LLM prompt. The paper does not report a sensitivity study, so practitioners may need extensive search to replicate the reported gains in new settings.

---

> ### Author Rebuttal · Authors · 2025-07-31
>
> We sincerely appreciate the reviewer’s detailed and constructive comments. Our response to these concerns and questions are provided below.
>
> **W1 Clarification of AUC improvement, interpretation results examined by domain expert**
>
> We first clarify the calculation of claimed AUC improvement. In the weather dataset, our method achieves the best AUC score 0.808, and the second-best AUC score from TimeCAP is 0.742. The percentage of improvement based on TimeCAP is (0.808-0.742)/0.742 = 8.9%. Similarly, the average AUC improvement is 2.2% (average F1 improvement 2.78%). We will differentiate the absolute improvement and relative percentage improvement in our manuscript.
>
> We appreciate the reviewer’s suggestion for expert evaluation. We would like to kindly provide our justification as follows. Firstly, our framework is primarily task-performance driven, where explanations serve to enhance prediction accuracy. Secondly, we focus on general-domain datasets including weather, finance and influenza surveillance (test-positive and mortality), where we believe the contexts are relatively accessible to general AI practitioners. The results of multi-modal prototypes and case-studies (Figures 3-4, 9-13) are generally relevant and understandable without requiring deep domain expertise. However, we agree with the reviewer that domain experts can provide a more rigorous evaluation of explanation clarity, and will involve them in the future work for specialized domains like clinical decision-making.
>
> **W1, W3, L2 Task expansion on other benchmarks**
>
> Thanks for suggesting additional tasks and benchmarks. We would also like to clarify that we still achieve noticeable improvement (a 6.62% relative RMSE improvement over the second-best baseline) in the regression setting on the Finance dataset. Overall, our method has been evaluated across three domains, four datasets, and two tasks. Nevertheless, we agree with the reviewer that extending the evaluation is valuable, and we plan to explore this in the future work.
>
> **W2 Computational cost and trade-off**
>
> We thank the reviewer for this practical concern and address them as follows. For a dataset with 5,000 samples (including 1,000 testing cases) with average length of 400 tokens (~300 words based on OpenAI tokenizer). Following the prompts and strategy detailed in Appendix D (Pages 22-24), we estimate the token cost per iteration over train/validation samples as: Prediction 3.80 M input tokens (output negligible); Reflection 1.66 M input / 0.04 M output; Refinement 4.28 M input / 1.60 M output. It yields ~9.74 M input and ~1.64 M output tokens per iteration.
>
> We next compute the time cost per iteration. With GPT‑4o (Input latency as TTFT ~0.45 s; output speed 156 tokens/s), prediction takes ~0.5 h, reflection ~0.08 h, and refinement ~3.35 h, totaling ~3.9 h serially. Using Gemini‑2.0‑Flash (TTFT ~0.3 s; 236 tokens/s) reduces this to ~2.6 h. Prediction and refinement can also be parallelized (e.g., 20 concurrent calls cut GPT‑4o’s iteration time to ~12 min). Note that at test time we apply the fixed reflection to refine the test texts and perform prediction, yielding a reasonable inference latency of ~3.5 s/sample for GPT‑4o, and ~2.3 s/sample Gemini‑2.0‑Flash.
>
> We agree with the reviewer that the trade-off between performance and cost is important in practice. As shown in Figure 5 and discussed in Section 4.5, 1-2 iterations are sufficient to achieve clear performance gains. In practice, the iterations can be controlled by monitoring performance improvements on validation data and stopping early if gains are below a threshold, thereby balancing accuracy with computational cost. Moreover, cost can be further reduced by selective refinement (skipping examples already predicted correctly in prior iterations), making our method more tractable. We will add a detailed discussion in our updated version.
>
> **W4 Parameter selection and generalization**
>
> We appreciate the reviewer’s comments for parameter selection and respond as follows. Firstly, we would like to emphasize that the weight for output fusion is simple yet effective for augmented prediction. The optimal weight is selected by evaluating the values from 0 to 1 on the validation logits, providing an efficient and training-free solution for generalization. For the sensitivity of case-based examples (with prototypes) fed to LLM, the evaluation is provided in Figure 7 of Appendix B (Pages 18-19), with discussions in lines 742-751, showing the advantage of more relevant case-based explanations. We also provide qualitative ablation for the encoder regarding regularization coefficients in Figure 6 of Appendix B.
>
> To quantitatively assess their sensitivity, we provide the evaluation results for the multi-modal encoder on the Weather dataset (based on the refined testing texts), where each parameter is selected from [0.1,0.3] with interval 0.05, following Appendix A.2 (Page 17). As shown in the below table, model performance remains relatively stable across different values. For both modalities, moderate values of $\lambda_1$ and$ \lambda_2$ tend to yield better performance, suggesting the clustering and evidencing effectively guide meaningful prototype learning. A slight diversity constraint ($\lambda_3$) also helps to keep a compact interpretation prototype space. We agree with the reviewer that a more detailed sensitivity evaluation is beneficial for new settings, and we will include an expanded analysis in the updated version.
>
> |Time|Metric|0.1|0.15|0.2|0.25|0.3|
> |-|-|-|-|-|-|-|
> |$\lambda_1$|F1|0.628±0.030|0.627±0.027|0.633±0.029|0.627±0.028|0.625±0.030|
> ||AUC|0.733±0.022|0.731±0.026|0.736±0.026|0.731±0.024|0.729±0.023|
> |$\lambda_2$|F1|0.632±0.028|0.634±0.028|0.627±0.032|0.626±0.028|0.628±0.028|
> ||AUC|0.734±0.024|0.736±0.022|0.733±0.028|0.729±0.021|0.733±0.025|
> |$\lambda_3$|F1|0.631±0.028|0.631±0.027|0.630±0.028|0.624±0.032|0.628±0.029|
> ||AUC|0.736±0.022|0.735±0.024|0.733±0.024|0.727±0.026|0.733±0.025|
>
> |Text|Metric|0.1|0.15|0.2|0.25|0.3|
> |-|-|-|-|-|-|-|
> |$\lambda_1$|F1|0.629±0.027|0.629±0.026|0.631±0.027|0.635±0.027|0.634±0.028|
> ||AUC|0.732±0.024|0.736±0.024|0.740±0.026|0.738±0.023|0.735±0.025|
> |$\lambda_2$|F1|0.632±0.031|0.631±0.026|0.635±0.024|0.629±0.026|0.630±0.028|
> ||AUC|0.734±0.027|0.736±0.021|0.740±0.024|0.735±0.021|0.736±0.028|
> |$\lambda_3$|F1|0.635±0.026|0.633±0.023|0.628±0.025|0.625±0.029|0.627±0.027|
> ||AUC|0.740±0.021|0.736±0.025|0.731±0.023|0.729±0.027|0.731±0.026|
>
> **Q1 Channel-based clustering for long-term time series predictions**
>
> Thanks for raising channel-based clustering in our discussion. Based on our knowledge, it assigns channel-wise embedding to forecasters and effectively exploits channel similarity via a clustering objective that enhances accuracy. However, its scope differs from our prototype design, thus may not serve as a replacement. Our design focuses on interpretable prediction via case-based reasoning over modalities, based on the explanations identified from training data. While the channel-based clustering provides some insights into channel groups, its granularity is different from our instance-level explanation. We recognize its potential value for multi-modal time series analysis, and will carefully discuss this work and related methods in the related work.
>
> **Q2 Segment-level prototype for long-term and irregular time series predictions**
>
> We would like to clarify that segment-level modeling remains effective for time series forecasting. The state-of-the-art models like PatchTST and tPatchGNN[1] also rely on patching that essentially operates at the segment level to handle long and irregular time series, respectively.
>
> Next, we sketch how to adapt our method to long-term forecasting. One solution is to increase the convolution window and stride (similar as patching), optionally with other mechanisms like multi-scale modeling or time series decomposition. Alternatively, a prototype layer with learning objectives can be added to the projection layer after patching in PatchTST. In this way, prototypes can capture periodic or salient patterns in long-term time series benchmarks.
>
> [1] Irregular Multivariate Time Series Forecasting: A Transformable Patching Graph Neural Networks Approach.
>
> **Q3 Additional information introduced by TimeXL in the case study**
>
> We provide a case study in Figure 4 where prediction is calibrated to a rainy outcome by TimeXL. As noted in lines 312-317, the refined texts preserve the statement of stability in original texts while emphasizing more on humidity and wind as more indicative of rain, guided by reflections from training examples, contributing to the calibrated prediction. The refined texts also improve the quality of learned prototypes by filtering out less relevant cues. “Gradual increase in temperature” becomes less relevant as more indicative phrases in the same input yield higher similarity scores, better highlighting upcoming rainy conditions. Nevertheless, we thank the reviewer’s advice and will provide more examples in our updated version.
>
> **L1 Evaluating text quality to decide if multi-modal method is needed**
>
> We sincerely thank the reviewer for pointing out this practical challenge. While our method handles noisy or redundant textual inputs through iterative and reflective feedback and supports interpretable predictions, we agree that its effectiveness may be limited when the texts are extremely sparse or uninformative. In such cases, it is important to assess the text quality for model development. Integrating domain-specific prompts and external knowledge retrieval could also help. We will discuss this issue in our updated version.
>
> **L3 Assessing the current benchmarks and LLM pretrain data**
>
> We thank the reviewer for raising this concern. The potential leakage regarding the LLM pretraining corpus remains a shared challenge in current LLM-based research, and we will carefully assess the risk in the future work.

---

> > ### Comment · Reviewer_Vxdr · 2025-08-06
> >
> > Thank you to the authors for the time you've spent on responses both to me and the other reviewers.
> >
> > In particular I believe the additional work around sensitivity analysis both on parameter selection for the balance of the regularization terms and on the prototype length in response to reviewer srsZ help strengthen the argument that the method is robust in various settings and providing guidance for selection in new environments.
> >
> > As for the responses to W1, I'm not entirely convinced by the claims that "we focus on general-domain datasets... where we believe the contexts are relatively accessible to general AI practitioners". I don't think this general knowledge claim is backed up in the paper or by any existing literature (though I'm open to being proven otherwise). I would suggest softening the claims  of explainability which are central to the title and novelty claims but not thoroughly backed-up. I agree that the framework is performance-driven and I think the research stands up there and so additional unverified claims about explainability are not required. Are there further changes that would be worth making here?

---

> > > ### Author Response · Authors · 2025-08-07
> > >
> > > We sincerely appreciate your recognition of our additional work and the robustness of our method across different settings.
> > >
> > > Regarding your concern on expert examination, we completely understand and acknowledge the importance of this point. We agree that our original phrasing around “general knowledge” could have been more carefully framed, and we appreciate your suggestion to clarify this aspect. To that end, we will revise the manuscript to clearly state that our primary focus is on prediction-oriented explanations designed to enhance performance.
> > >
> > > We also appreciate your insight into the value of incorporating domain expert feedback. Building on our previous response, we will include a discussion in the revised version that acknowledges this challenge and outlines directions for integrating expert evaluation into future work. Thank you again for your thoughtful and constructive comments. They are truly helpful in strengthening our paper.

---

### Official Review · Reviewer_srsZ · 2025-07-02

**Clarity:** 3
**Significance:** 3
**Originality:** 3
**Rating:** 4
**Confidence:** 3

**Summary:**

This paper introduces TimeXL, a novel multi-modal time series prediction framework that integrates a prototype-based encoder with a closed-loop system of three Large Language Models (LLMs): prediction, reflection, and refinement. The approach enhances both predictive accuracy and interpretability by iteratively improving explanations and model outputs through LLM-driven feedback. TimeXL was evaluated on four real-world datasets spanning weather, finance, and healthcare, showing improvements in AUC by up to 8.9% compared to state-of-the-art baselines.

**Questions:**

where are the initial time series prototypes and text prototypes from? Random initialization?

**Ethical Concerns:**

["NO or VERY MINOR ethics concerns only"]

**Final Justification:**

I keep the "borderline accept" rating.

**Quality:**

3

**Strengths And Weaknesses:**

Strengths
1.	The framework, which combines the prototype-based encoder and iterative LLM agents (predict-reflect-refine), is well-motivated.
2.	Strong empirical results and Explainability: Demonstrates significant performance improvements across diverse datasets, highlighting robustness. Case-based reasoning with multi-modal prototypes offers transparent and human-understandable predictions.
3.	Comprehensive evaluation: Includes ablation studies, qualitative case studies, and comparisons with numerous baselines.
Weaknesses
1. Computational cost: Running multiple LLMs iteratively may be resource-intensive, which might limit real-time deployment.
2. Ambiguity in prototype learning: The clarity of how well the prototypes generalize across different tasks or domains is underexplored.
3. Unreliable on time series regression. This method heavily relies on time series segmentation, which may introduce bias based on arbitrary windowing choices. Prediction LLM returns “increase”, “decrease”, or “neutral” for prediction, which is unreasonable for a long-horizon time series regression, where granularity and uncertainty quantification are essential.
4. No sensitivity analysis to the subsequence length.

---

> ### Author Rebuttal · Authors · 2025-07-31
>
> We sincerely appreciate the reviewer’s constructive comments. Our responses to the concerns are listed below.
>
> **W1 Computational cost for real-time deployment**
>
> We thank the reviewer for pointing out this practical concern and address it as follows.
>
> First, we provide an example demonstrating the computational cost of three LLMs in the iterative process. Following the prompts and strategy detailed in Appendix D (Pages 22-24), each iteration over  4000 training and validation samples (average length of 400 tokens or ~300 words) requires ~9.74 M input tokens (Prediction LLM 3.80 M + Reflection LLM 1.66 M + Refinement LLM 4.28 M) and yields ~1.64 M output tokens. With GPT‑4o (with input latency as TTFT ~0.45 s and output speed ~156 tokens/s), it corresponds to ~3.9 hours serially (prediction takes ~0.5 h, reflection ~0.08 h, and refinement ~3.35 h). The prediction and refinement steps can also be parallelized, reducing time cost per iteration to ~12 min with 20 concurrent calls. For a more efficient variant Gemini-Flash-2.0 (evaluated in Appendix B, Pages 18-19) with TTFT ~0.3 s and ~236 output tokens/s, the iteration time can be further reduced to ~8 min.
>
> We would like to emphasize that we do not run reflection in the testing/deployment stage, but instead apply the fixed reflection selected during validation to refine the test texts (as noted in lines 245-250), reducing inference cost and latency. Based on our prompt lengths, the refinement and prediction steps take ~3.46 s per sample using GPT‑4o, and ~2.29 s per sample for Gemini-2.0-Flash, which is generally reasonable for real-time deployment.
>
> Second, we would like to discuss more strategies for reducing computational cost in the iterative process. As shown in Figure 5 and discussed in Section 4.5, 1-2 iterations are often sufficient to achieve clear performance improvements. In real-world applications, the number of iterations can be adaptively controlled to balance performance gains and cost. For example, one can monitor performance gains on validation data after each iteration and stop early if the improvement falls below a predefined threshold. To further reduce the cost, we can also do selective refinement by skipping training examples that were already predicted accurately in previous iterations. These strategies help make our method more computationally tractable.  We will include an extended discussion in our updated version.
>
>
> **W2 Clarity regarding generalization of prototype across tasks and domains**
>
> We would like to clarify that we have evaluated the learned prototypes across four datasets and two tasks.  Besides the multi-modal prototypes (Figure 3) and the case-based reasoning example (Figure 4) on the Weather dataset in Section 4.3-4.4, we also provided additional results for all datasets, including multi-modal prototypes and examples in Figures 9-13 of Appendix C (Pages 20-22). These results demonstrate that our prototypes capture meaningful and context-specific patterns across diverse domains. Beyond classification, we also evaluated the regression setting on the Finance dataset. This includes task-specific design, performance evaluation, prediction visualization, ablation study, iterative analysis, and case study, which are presented in Tables 6-8 and Figures 24-27 in Appendix F (Pages 26-29).
>
> **W3 Time series segmentation for long-term times series regression**
>
> Thanks for raising this concern. We would like to clarify that time series segmentation is an effective strategy for time series regression, as the state-of-the-art forecasters PatchTST and TimeLLM, also rely on patching operations (essentially segmentation). In the regression setting presented in Appendix F (Pages 26-29), we also let prediction LLM output a numerical value in addition to the trend outcome (prompt in Figure 25). While multi-modal long-term time series prediction is not our primary focus, we would like to elaborate how our method can be adapted to this setting. One solution is to perform the single-step prediction and iterate over the forecast horizons, which provides fine-grained information for analysis. It is also possible to use a larger context window and let LLM generate multi-steps predictions in one-shot, while using the trend classification over window statistics as an auxiliary guidance signal to enhance temporal reasoning. As noted in Appendix H (Page 30), we will explore this setting with more viable solutions in our future work.
>
> **W4 Sensitivity analysis of prototype (subsequence) length**
>
> We thank the reviewer for pointing this out. To assess the sensitivity of prototype (subsequence) length, we conducted additional experiments on the multi-modal encoder by varying the lengths for both time series and text modalities across the Weather and Finance datasets (based on the refined testing texts). The results of their combinations and summarized statistics are presented below.
>
> We observe that performance is relatively stable across a range of prototype lengths in the Weather dataset, with optimal results typically achieved at moderate lengths for both modalities. For time series, the 8-hour segments are often sufficient to capture the typical temporal patterns of weather conditions, and the 12-token window (where we used BERT for text embedding, as indicated in line 280) provides enough local contexts for weather prediction. For the Finance dataset, which operates on a business-day granularity, performance improves with longer lengths of time series prototypes that capture important financial trends. For the textual modality, we use Sentence-BERT to embed financial news at the half-sentence granularity (as noted in line 324), and shorter text windows (1-2 segments) typically provide compact and indicative market conditions for prototypes, making them effective for trend prediction.
>
> These results suggest that our model is relatively robust to the choice of prototype length within a reasonable range that aligns with the temporal characteristics of the data. We will include an extended analysis in our updated manuscript.
>
> Weather Dataset
> ||||||||||||||||||
> |-|-|-|-|-|-|-|-|-|-|-|-|-|-|-|-|-|
> |Time Series|4|4|4|4|8|8|8|8|12|12|12|12|16|16|16|16|
> |Text|4|8|12|16|4|8|12|16|4|8|12|16|4|8|12|16|
> |F1|0.605|0.644|0.650|0.627|0.640|0.633|0.659|0.635|0.613|0.614|0.641|0.633|0.618|0.612|0.639|0.614|
> |AUC|0.731|0.754|0.758|0.749|0.755|0.746|0.763|0.753|0.741|0.739|0.755|0.750|0.743|0.737|0.753|0.738|
>
> |Time Series - Overall|F1|AUC|
> |---|---|---|
> |4|0.632 ± 0.018|0.748 ± 0.011|
> |8|0.642 ± 0.011|0.754 ± 0.008|
> |12|0.625 ± 0.013|0.746 ± 0.007|
> |16|0.621 ± 0.012|0.743 ± 0.007|
>
> |Text - Overall|F1|AUC|
> |---|---|---|
> |4|0.619 ± 0.014|0.743 ± 0.008|
> |8|0.626 ± 0.015|0.744 ± 0.008|
> |12|0.647 ± 0.008|0.757 ± 0.004|
> |16|0.627 ± 0.009|0.748 ± 0.007|
>
>
> Finance Dataset
>
> |||||||||||||||||
> |-|-|-|-|-|-|-|-|-|-|-|-|-|-|-|-|
> |Time Series|1|1|1|2|2|2|3|3|3|4|4|4|5|5|5|
> |Text|1|2|3|1|2|3|1|2|3|1|2|3|1|2|3|
> |F1|0.428|0.433|0.429|0.476|0.462|0.424|0.484|0.470|0.471|0.538|0.529|0.510|0.616|0.588|0.559|
> |AUC|0.592|0.597|0.595|0.655|0.630|0.605|0.671|0.658|0.658|0.722|0.712|0.708|0.788|0.767|0.753|
>
>
> |Time Series - Overall|F1|AUC|
> |---|---|---|
> |1|0.430 ± 0.002|0.595 ± 0.002|
> |2|0.454 ± 0.028|0.630 ± 0.021|
> |3|0.475 ± 0.006|0.662 ± 0.006|
> |4|0.526 ± 0.014|0.714 ± 0.006|
> |5|0.588 ± 0.029|0.769 ± 0.018|
>
>
> |Text - Overall|F1|AUC|
> |---|---|---|
> |1|0.508 ± 0.073|0.666 ± 0.074|
> |2|0.496 ± 0.061|0.673 ± 0.067|
> |3|0.458 ± 0.045|0.664 ± 0.056|
>
>
> **Q1 Initial time series and text prototypes**
>
> Thanks for this question! Yes, we introduce time series and text prototypes as randomly initialized training parameters, which are projected to an initial sample before training, where the specific implementation is provided in the supplementary code.

---

> > ### Comment · Reviewer_srsZ · 2025-08-06
> >
> > Thanks for the details. I will keep my score.

---

### Official Review · Reviewer_83We · 2025-07-03

**Clarity:** 2
**Significance:** 3
**Originality:** 3
**Rating:** 4
**Confidence:** 3

**Summary:**

The authors built a closed-loop prediction framework, consisting of a time-series encoder to provide initial predictions and explanations, a prediction LLM to update predictions through reasoning, and a reflection LLM to compare the results with the ground truth. This workflow is multimodal, allowing for time-series and textual inputs, and provides interpretable explanations. This setup was tested on 4 real-world datasets and showed 8.9% AUC improvement.

**Questions:**

* How did the authors choose to use GPT 4o, Gemini-2.0-Flash and GPT-4o-mini over other LLMs? What was the performance across other models?
* Could the authors explain why a reflection LLM is needed? Especially compared to just using a metric to evaluate the differences?
* What’s the computational overload for this closed loop? When do you stop looping? How do you determine this stop?
* Discussion on broader impact and limitations should be put in the main text
* Adding a qualitative evaluation of faithfulness from domain experts would support the downstream utility of this tool

**Ethical Concerns:**

["NO or VERY MINOR ethics concerns only"]

**Final Justification:**

Thank you to the authors for addressing my comments. I will keep my score, subject to the revisions being added to the final paper.

**Limitations:**

* There are no limitations noted in the paper nor future directions.
* Novelty regarding methodology is low since the main contribution is combining and prompting GPT/Gemini models
* The authors mention their contribution as “deliver both accurate and interpretable multi-modal forecasting” but yet they do not show experiments looking at forecasting in the main text (rather focused on classification tasks)

**Paper Formatting Concerns:**

None noticed

**Quality:**

3

**Strengths And Weaknesses:**

Strengths
* Paper is overall clear and easy to read
* Addresses gap in interpretability of multimodal LLMs, which is an important and relevant challenge
* The iterative closed-loop workflow allows for continued performance enhancement
* Provides human-interpretable explanations from multiple modalities
* Multiple ablation studies, real-world experiments and baselines helps justify the utility and generalizability

Weakness:
* Computational overload for the closed loop may be an issue depending on stop function
* No discussion of results, goes straight from results to short conclusion
* No qualitative evaluation of faithfulness/utility

---

> ### Author Rebuttal · Authors · 2025-07-31
>
> We sincerely appreciate the reviewer’s insightful feedback. Below we provide our responses to the concerns and questions.
>
> **W1 and Q3: Computation cost for closed loop optimization**
>
> We sincerely thank the reviewer for raising this practical concern and provide our responses as follows.
> We first quantitatively discuss the computation cost of LLMs in the closed loop, considering a dataset with 5,000 samples (4,000 for training and validation), and average length of 400 tokens (~300 words based on OpenAI tokenizer). Following the prompts and strategy detailed in Appendix D (Pages 22-24), we calculate the token usage for each LLM per call and per iteration (configured with 10 × 20‑token prototypes for Prediction LLM and a batch size of 50 for Reflection LLM).
>
> (1) Tokens per call for each LLM: the Prediction LLM uses ~950 input tokens per sample (output negligible); the Reflection LLM uses ~20,730 input tokens per batch and ~500 output tokens; and the Refinement LLM uses ~970 input tokens and fewer than 400 output tokens.
>
> (2) Token cost per iteration (4,000 training and validation samples): Prediction takes ~3.80 M input tokens (output negligible), Reflection ~1.66 M input and ~0.04 M output tokens, and Refinement ~4.28 M input and ~1.60 M output tokens. Total: 9.74 M input tokens and 1.64 M output tokens per iteration.
>
> We now calculate the time cost per iteration. Using GPT‑4o (Input latency as TTFT ~ 0.45 s per call; Output speed ~156 tokens/sec), prediction takes 0.50 h, reflection 0.08 h, and refinement 3.35 h, yielding ~3.9 h serially in total. Note that the prediction and refinement steps can be parallelized; for example, 20 concurrent calls reduce GPT‑4o’s iteration time to ~12 min. Using Gemini-2.0-Flash (TTFT ~ 0.30 s, ~236 output  tokens/sec) further reduces it to ~8 min.
>
> We also provide the test-time latency as follows. Note that we apply the fixed reflection selected during validation to refine the testing texts (as noted in lines 245-250), reducing inference cost and latency. For GPT‑4o, the refinement and prediction steps take ~3.46 s per sample, and for Gemini-2.0-Flash, ~2.29 s per sample. In general, the computation cost per iteration and testing latency are reasonable.
>
> Next, we elaborate the iteration strategy. As shown in Figure 5 and discussed in Section 4.5, 1-2 iterations are sufficient to see clear performance improvements. In our study, we set a fixed number of iterations to explore the further potential of improvement from iterative reflective reasoning on texts. In real-world applications, the number of iterations can be adaptively controlled by balancing performance gains against computational budget. For example, one may monitor the performance improvement (or its average) after each iteration on validation data, and terminate the loop early if it is below a predefined threshold. To further reduce the cost, we can also do selective refinement by skipping training examples that were already predicted accurately in previous iterations. These strategies help mitigate the issue of computational cost, making our method more tractable.  We will provide a detailed discussion in the appendix.
>
> **W2: Lack of results discussions**
>
> We would like to kindly clarify that we provided discussions for each of the results presented, including key observations and how they validate our proposed components.
>
> In Section 4.3, we discussed the real-world implications of multi-modal prototypes across both rain and not rain classes on weather datasets, showing these prototypes capture key contexts for the following case-based explanations. In Section 4.4, we further provided a case study and elaborated how the rendered multi-modal prototypes highlight contextual information linked to the upcoming weather event through the refinement process. More results are discussed in Appendix C (Pages 20-22).
>
> To further validate the workflow, we provided and discussed iterative analysis in Section 4.5 and ablation study in Section 4.6 (More in Appendix B, Pages 18-19). We specifically discussed the performance improvement behavior through iterative reflective refinement, as well as the benefit of integrating time series and textual modalities, the effectiveness of explainable artifacts regarding providing relevant contextual guidance, and the advantage of augmented prediction between multi-modal encoder and LLM.
>
> **W3 and Q5: Qualitative evaluation of faithfulness from domain expert**
>
> We appreciate the reviewer’s helpful suggestion and kindly provide our justifications as follows. Firstly, our current framework is primarily task-performance driven, where explanations serve to facilitate accurate predictions. Moreover, we focus on general-domain datasets including weather, finance and influenza surveillance (domain and task descriptions are provided in lines 253-266, Section 4.1), where the contexts are relatively accessible to AI practitioners. The explanations including multi-modal prototypes and case studies (provided in Figures 3,4, 9-13) are relevant and generally intuitive without requiring deep domain expertise. Nevertheless, we acknowledge the value of involving domain experts to ensure the clarity and usefulness of explanations, and we agree that qualitative evaluation from domain experts would further support the downstream utility. We will consider domain expert evaluation in specialized applications like clinical narratives for future work.
>
> **Q1: Choices of LLMs**
>
> We appreciate the reviewer’s interest in our choice of LLMs. Our framework is designed to be model-agnostic and can accommodate LLMs with sufficient context reasoning capabilities. As such, we selected GPT‑4o, GPT‑4o-mini, and Gemini-2.0-Flash based on their strong reasoning and inference  performance on diverse benchmarks, their diversity in model scale and architecture, and their accessibility at the time of our experiments.
>
> Specifically, GPT-4o was selected for its strong general-purpose reasoning. GPT-4o-mini and Gemini-2.0-Flash were included as more cost-efficient variants to assess the framework’s performance under limited computation and model capacity, within and beyond OpenAI. We provided performance evaluation and iterative study, with corresponding discussions in Appendix B (Pages 18-19). The results show how LLM capability influences TimeXL in understanding real-world context, and validate its consistent effectiveness across different LLM base models.
>
> **Q2: The necessity of reflection LLM**
>
> We appreciate the reviewer’s question. The key reason for using a reflection LLM rather than relying solely on scalar evaluation metrics is to enable generalization to the testing phase, where ground truth labels are unavailable to guide textual refinement.
>
> During training, the reflection LLM compares predicted values against ground truth labels to identify issues such as textual noise or misalignment. This reflection process, detailed in our prompting strategy (lines 792–800) and template designs (Figures 16-18, Appendix D, Pages 22-24), generates targeted feedback that is used to refine the input text.
>
> At test time, since ground truth labels are not available, metrics cannot provide usable refinement signals. Instead, we apply the reflection strategy selected based on validation performance (Section 3.3.2 and Algorithm 1) to refine test samples. This mimics how a trained model generalizes to unseen data and enables label-free inference-time improvements.
>
> Moreover, compared to scalar metrics (e.g., accuracy, distance), the reflection LLM provides richer, context-aware feedback by reasoning over the input text, model prediction, and training outcomes. This leads to more meaningful and interpretable refinements. Examples of such reflection reports guiding refinement are shown in Figures 20-23 (Appendix E, Pages 25-26).
>
> **Q4 and L1: Broader impact, limitation and future directions**
>
> We thank the reviewer for this suggestion. We acknowledge the importance of clearly articulating the broader impact and limitations. While these discussions are currently placed in Appendix H (limitations and future work, Page 30) and Appendix I (broader impact, Page 31), we agree that it could be better to incorporate them into the main text of the paper. We will appropriately integrate these sections in the updated version.
>
> **L2 Method novelty**
>
> We appreciate the reviewer’s comment and would like to clarify the novelty of our method as follows. Our core contribution lies in a structured and iterative synergy between a multi-modal time series encoder and multiple role-specific LLMs, designed to improve both predictive accuracy and interpretability. This end-to-end reflective refinement workflow, including error diagnosis and correction, moves beyond traditional prompting. We designed tailored prompt templates and modular reasoning flows (Appendix D, Pages 22-24), and validated them across multiple tasks and datasets, showing that our synergistic design goes beyond simple prompting or model combination.
>
> **L3 Experiments of forecasting**
>
> We appreciate the reviewer’s observation and would like to clarify as follows. As noted in lines 129-132, we focus on discrete prediction settings to reflect real-world multi-modal decision-making scenarios, and we demonstrated regression-based forecasting through experiments on Finance dataset in Appendix F (Pages 26-29). This includes a model and prompt design tailored for regression tasks (Figures 24-25), along with performance evaluation against state-of-the-art forecasting baselines (Table 6), regression visualizations (Figure 26). We also provide an ablation study, iterative refinement analysis, and a case study (Tables 7-8, Figure 27). We chose to include these results in the appendix due to space limitations, but we believe they clearly support our claim of delivering both accurate and interpretable multi-modal forecasting. We will try to highlight these results more prominently in the main text.

---

> ### Author Response · Authors · 2025-08-08
>
> Dear Reviewer 83We,
>
> We sincerely appreciate your thoughtful and constructive feedback, as well as the recognition of our work. We have made our best effort to address the raised concerns and questions. Considering the limited time in the author-reviewer discussion period, we would be grateful if you could let us know whether our responses have adequately addressed your concerns.

---

> ### Comment · Reviewer_83We · 2025-08-09
>
> Thank you to the authors for clarifying and addressing my questions. I will keep my score

---

### Decision · Program_Chairs · 2025-09-17

**Decision:**

Accept (poster)

**Comment:**

Following the final round of reviews, the paper received four borderline accept decisions. The reviewers recognized the technical novelty of the proposed approach and noted its strong performance across the evaluated benchmarks. They also emphasized that the explainability feature represents a significant contribution, addressing an important gap in the field of explainable AI for time series analysis. However, the reviewers also raised concerns regarding the computational overhead associated with the method, suggesting this may limit its scalability or applicability to other domains. The rebuttal addressed some of these concerns, but only partially satisfied the reviewers’ concerns.